# Uncertainty in non-stationary frequency analysis of Korea's daily rainfall POT excesses associated with covariates

Okjeong Lee

Department of Environmental Engineering, Pukyong National University, Busan 48513 Korea

Jeonghyeon Choi

Division of Earth Environmental System Science, Major of Environmental Engineering, Pukyong National University, Busan 48513 Korea

Jeongeun Won

Division of Earth Environmental System Science, Major of Environmental Engineering, Pukyong National University, Busan 48513 Korea

Sangdan Kim

Corresponding author, Department of Environmental Engineering, Pukyong National University, Busan 48513 Korea. E-mail: skim@pknu.ac.kr

submitted at Hydrology and Earth System Sciences

**Abstract**

Several methods have been proposed to analyze the frequency of non-stationary anomalies. The applicability of the non-stationary frequency analysis has been mainly evaluated based on the agreement between the time series data and the applied probability distribution. However, since the uncertainty in the parameter estimate of the probability distribution is the main source of uncertainty in frequency analysis, the uncertainty in the correspondence between samples and probability distribution is inevitably large. In this study, an extreme rainfall frequency analysis is performed that fits the Peak-over-threshold series to the covariate-based non-stationary Generalized Pareto distribution. By quantitatively evaluating the uncertainty of daily rainfall quantile estimates at 13 sites of the Korea Meteorological Administration using the Bayesian approach, we tried to evaluate the applicability of the non-stationary frequency analysis with a focus on uncertainty. The results indicated that the inclusion of dew-point temperature (DPT) or surface air temperature (SAT) generally improved the goodness of fit of the model for the observed samples. The uncertainty of the estimated rainfall quantiles was evaluated by the confidence interval of the ensemble generated by the Markov chain Monte Carlo. The results showed that the width of the confidence interval of quantiles could be greatly amplified due to extreme values of the covariate. In order to compensate for the weakness of the non-stationary model exposed by uncertainty, a method of specifying a reference value of a covariate corresponding to a non-exceedance probability has been proposed. The results of the study revealed that the reference co-variate plays an important role in the reliability of the non-stationary model. In addition, when the reference co-variate was given, it was confirmed that the uncertainty reduction of quantile estimates for the increase in the sample size was more pronounced in the non-stationary model. Finally, it was discussed how information on global temperature rise could be integrated with DPT or SAT-based non-stationary frequency analysis. It has been formulated how to quantify the uncertainty of the rate of change in future quantile due to global warming using rainfall quantile ensembles obtained in the uncertainty analysis process.

Keywords: Co-variate, Generalized Pareto distribution, Non-stationary frequency analysis, Peak-over-threshold time series, Uncertainty

## 1. Introduction

Human activity in the last century has caused global surface air temperature to rise (Karl et al., 2009; Min et al., 2011). When the temperature rises by 1 ℃, the moisture retention capacity in the atmosphere increases by about 7 %, which directly affects precipitation (Trenberth, 2011; Sim et al., 2019). The higher the water vapor in the atmosphere, the more likely it is to increase precipitation (Berg et al., 2013), and increasing surface air temperature and increasing atmospheric moisture content can increase probable maximum precipitations or rainfall extremes (Kunkel et al., 2013; Lee and Kim, 2018). As a result, global warming damages the performance of drainage system infrastructure such as embankments, sewers and dams (Das et al., 2011; Jongman et al., 2014), increasing the risk of climate extremes (Emori and Brown, 2005; Hao et al., 2013). In fact, looking at ground observations around the world shows that rainfall extremes have increased significantly over the past century (Karl and Knight, 1998; DeGaetano, 2009). Global studies have shown that precipitation has increased in northern Australia, Central Africa, Central America, and Southwest Asia (Groisman et al., 2012).

The current infrastructure design concept for dealing with rainfall extremes is based on the estimation of design rainfall depth using frequency analysis of annual maximum series for various durations in a region (Madsen et al., 2002; Madsen et al., 2009; Hosking and Wallis, 2005; Sugahara et al., 2009; Haddad et al., 2011; Willems, 2013; Kim et al., 2020). Current design rainfall depth is based on the concept of stationarity in time, which assumes that the probability of occurrence of extreme rainfall events is not expected to change significantly over time. However, natural environmental changes, such as global warming, have a serious impact on the assumptions of the stationarity of the observations. Non-stationarity is an important issue that can never be ignored in areas related to drainage system design, as it can alter the design flood volume obtained using the stationary frequency analysis of observed rainfall extremes. The probability of occurrence of extreme rainfall events is expected to change due to global warming (Lee et al., 2016), and this change is called non-stationarity by many authors (Alexander et al., 2006; Gregersen et al., 2013).

Several methods have been proposed to address non-stationarities in the time series (Cunha et al., 2011; Yilmaz et al., 2013; Jang et al., 2015, Moon et al., 2016), and many studies have been conducted to examine changes in design rainfall depth or return levels under non-

stationary conditions (Salvadori and DeMichele, 2010; Graler et al., 2013; Hassanzadeh et al., 2013; Salas and Obeysekera, 2013; Shin et al., 2014; Choi et al., 2019). Looking at the probability distributions and parameters applied to the above studies, most of the non-stationary frequency analysis is performed by expressing specific parameters of the Gumbel or Generalized Extreme Value (GEV) distribution as a function of covariate including time (Kim et al., 2017). In extreme rainfall series, non-stationarity may be explicitly expressed as a function of time, but may also be related to climate variables in the same or preceding time periods where rainfall extremes occurred (Zhang et al., 2010). Several studies have reported that it was reasonable to use climate variables rather than time for covariates to represent non-stationarities in the non-stationary frequency analysis (Agilan and Umamahesh, 2016; Sen et al., 2020). Recently, studies have been performed that analyze the non-stationary frequency using climate variables for annual maximum rainfall series (Villarini et al., 2012; Agilan and Umamahesh, 2017; Lee et al., 2018; Ouarda et al., 2019). In addition, studies have been conducted to analyze the non-stationary frequency using Peak-over-threshold (POT) series for the purpose of reducing the uncertainty occurring in the sample size (Tramblay et al., 2013; Jung et al., 2018; Lee et al., 2020).

In this research trend, what is of interest in this study is how to examine the relative superiority of the stationary and non-stationary models. Most studies use Akaike Information Criterion (AIC) and similar indicators, which evaluates how well the time series and probability distribution match, to select the optimal model from various candidate non-stationary model, including the stationary model (Akaike, 1974; Ganguli and Coulibaly, 2017; Iliopoulou et al., 2018; Lee et al., 2020). However, the results of selecting the optimal model by these methods are highly likely to vary depending on the sample size. Efforts to develop and apply a non-stationary model for frequency analysis to reflect changing environmental conditions can be frustrated by the additional uncertainty associated with the model's complexity, working with sampling uncertainty. In other words, the reliability of rainfall quantiles estimated by a complex non-stationary model may not be substantially improved, or when various environmental conditions are reflected, insufficient model reliability can easily lead to physically inconsistent results (Serinaldi and Kilsby, 2015). From this point of view, investigating which model has less uncertainty in rainfall quantile as a result of frequency analysis can be an important determinant in selecting an optimal model. This is because a model with a relatively smaller

uncertainty in the estimated rainfall quantile can be regarded as a more reliable model.

Whether or not the non-stationary model provides more reliable rainfall quantile estimates than the stationary model raises a lot of controversy. Serinaldi and Kilsby (2015) warned that uncertainty in non-stationary models might be greater since non-stationary models were more complex than stationary models. Agilan and Umamahesh (2018) investigated the effect of covariate selection on uncertainty in the covariate-based non-stationary analysis using annual maximum series. Ouarda et al. (2020) indicated that uncertainty was likely to work as a major weakness in the applicability of the non-stationary model through the analysis of UAE annual maximum rainfall series.

In this study, a non-stationary frequency analysis using dew point temperature (DPT) or surface air temperature (SAT) as a covariate is performed. As can be seen from Leopore et al. (2014), there is a strong scaling relationship between rainfall extreme and DPT or rainfall extreme and SAT. In addition, changes in DPT and SAT can directly affect the atmospheric moisture retention governed by the Clausius-Clapeyron equation, and in warmer climates, the moisture content of the atmosphere increases and the intensity of precipitation increases at a similar rate (Trenberth et. al., 2003; Giorgi et al., 2019). That is, according to the Clausius-Clapeyron relationship, the amount of moisture in the atmosphere increases exponentially as the temperature increases, and the amount of moisture in the atmosphere represents an increase rate of 6 - 7 %/K when other atmospheric conditions are kept constant. To obtain a necessary understanding of the relationship between daily rainfall and DPT and daily rainfall and SAT in Korea, two prior studies have been conducted (Sim et al., 2019; Lee et al., 2020). Sim et al. (2019) analyzed the effects of DPT and SAT on daily rainfall extremes. Their results indicated that even if there was some cooling effect in the event of summer rainfall (Ali and Mishra, 2017), daily rainfall extremes in Korea were very sensitive to DPT and SAT. Lee et al. (2020) presented a procedure for performing non-stationary frequency analysis using DPT or SAT as a covariate. They revealed that non-stationary frequency analysis using future DPT or SAT could yield more reasonable and persuasive projections of future rainfall extremes. The purpose of this study is to focus on the uncertainty of covariate-based non-stationary frequency analysis using DPT or SAT. The uncertainty in analyzing the non-stationary frequency of rainfall extremes using the annual maximum series inevitably includes the uncertainty due to the limitation of the sample size. In this study, the POT series is extracted from daily rainfall data

with the aim of reducing the uncertainty that comes from sample size as much as possible. Using the Bayesian approach, the parameters of the stationary and non-stationary Generalized Pareto (GP) distributions for the POT excesses are sampled from the posterior distribution. Using this, the performance of the stationary and non-stationary frequency analysis is investigated in terms of uncertainty. We will also examine how uncertainty in the non-stationary frequency analysis can be reduced by determining the appropriate covariate value (i.e., DPT or SAT value) corresponding to the rainfall quantile. Finally, the rate of change in rainfall quantile estimates for various DPT or SAT rise scenarios considering global warming will be analyzed based on uncertainty analysis.

## 2. Methods

### 2.1 Peak-over-threshold series and Generalized Pareto distribution

In this study, daily precipitation, daily DPT, and daily SAT data were used from 1961 to 2017 at 13 sites, including Busan and Seoul sites of the Korea Meteorological Administration (see Figure S1 of Supplementary Material). Figure 1 shows the results of quantile regression using daily precipitation data and DPT data on the day of precipitation observed at Busan and Seoul sites. Since the Korea Meteorological Administration only recognizes precipitation recorded at 0.1 mm or more per day as official precipitation, daily rainfall depth of 0.1 mm or more was applied to the analysis in this study. An example of this wet threshold can also be found in Chan et al. (2016) and Roderick et al. (2020). In fact, the application of a wet threshold does not significantly affect the results of quantile regression. A regression slope of 95 % extreme daily rainfall depth corresponding to DPT was estimated. For reference, the quantile regression equation for the quantile $\tau$ (0.95 in Figure 1) given in the quantile regression analysis is as follows:

$$ln\, R_\tau = a + bT, \tag{1}$$

where $R_\tau$ is the daily rainfall depth, and $T$ is the DPT of the day when the daily rainfall occurred. The following Eq. (2) was constructed using Eq. (1) to see how much the daily rainfall increases or decreases when DPT increases by 1 °C:

$$dR_\tau/K\, =\, 100(e^b - 1). \tag{2}$$

From Figure 1, it can be found that when DPT increases by 1 ℃, daily rainfall increases by 7 to 8 %.

[Figure 1. Sensitivity of 95 % daily rainfall depth to dew-point temperature at (a) Busan and (b) Seoul sites.]

In general, rainfall frequency analysis is performed using the annual maximum series or POT series. In the annual maximum series approach, the annual maximum rainfall series is generally assumed to follow the GEV distribution, and various studies have been conducted (Cheng et al., 2014). However, as the annual maximum series approach only considers one sample per year, the information contained in other data is completely ignored, so the POT approach to select the maximum number of samples for frequency analysis is being studied as an alternative (Hosseinzadehtalaei et al., 2017). In other words, since the POT approach uses more samples to enable accurate parameter estimation of the distribution, several studies recommend using the POT series instead of the annual maximum series (Yilmaz et al., 2014). The POT series is generally assumed to follow the GP distribution (Coles et al., 2001).

The cumulative probability distribution function of the stationary GP distribution for the POT series is as follows (Hosking and Wallis, 1987):

$$F(x) = 1 - \left(1 - k\frac{x - x_o}{\alpha}\right)^{1/k}, \tag{3}$$

where the range of $x$ is $x_o < x < \infty$, $\alpha$ is the scale parameter, and $k$ is the shape parameter ($k < 0$). The threshold $x_o$ should be determined in advance. The random variable $x$ has a value greater than $x_o$, and it is assumed that the occurrence of $x$ follows the Poisson distribution described by the annual incidence $\lambda$. The annual incidence $\lambda$ can be defined as the number of selected POT excesses divided by the observation year.

To ensure the independence of POT excesses, data larger than $x_o$ should be set so that they are not continuously selected. To ensure this, many studies have performed various clustering processes based on the time interval between extreme events (Gregersen et al., 2017). In this study, individual rainfall events were first separated from the daily rainfall

series. The applied Inter-Event Time Definition (IETD) is 1-day (Kim and Han, 2010). Then, in a rainfall event, it was set to select only one value at most as a POT series. For reference, in this study, the threshold $x_o$ for extracting POT excesses was assumed to be constant.

In non-stationary frequency analysis, temporally changing parameters are applied to the probability distribution function (PDF). Various types of functions are applied to the parameters that change over time. In general, the shape parameter is often set to constant (Lopez and Frances, 2013), but location or scale parameters are often considered functions of time or covariate. Ali and Mishra (2017) applied covariate to the location parameter of GEV, and Agilan and Umamahesh (2017) applied covariate to location and scale parameters of GEV. Non-stationary features in GP distribution are generally implemented by the scale parameter (Coles, 2001; Khaliq et al., 2006). Although non-stationarity can be expressed using the shape parameter, it is not a common practice since it is difficult to estimate the shape parameter, especially when considering covariates (Renard et al., 2006; Pujol et al., 2007). Although studies considering the non-stationarity of the threshold of the POT series have been conducted (Tramblay et al., 2012), in this study, the non-stationarity was given only to the scale parameter of the GP distribution as follows (Um et al., 2017):

$$\alpha_i = e^{\alpha_1 + \alpha_2 Z_i}, \tag{4}$$

where $i$ is the order of occurrence of POT excesses (1 to $n$), and covariate $Z_i$ is the climate variable corresponding to POT excesses (DPT or SAT on the day of POT excesses in this study). Eq. (4) tells how the covariate DPT or SAT is included in the model. The daily averaged DPT or SAT observed on the day of occurrence of each POT excess is included in the scale parameter of the GP distribution as shown in Eq. (4) to construct the non-stationary GP distribution. That is, when $\alpha_2 > 0$, the larger the DPT or SAT, the larger the scale parameter. Therefore, the parameters of the stationary GP distribution to be estimated are $\alpha$ and $k$, and the parameters of the non-stationary GP distribution are $\alpha_1$, $\alpha_2$, and $k$.

The formula for rainfall quantile $X_T$ corresponding to the return level of T-year in the non-stationary GP distribution using covariate is as follows:

$$X_T = x_o + \frac{1}{k} e^{\alpha_1 + \alpha_2 Z} \left[ 1 - \left( \frac{1}{\lambda T} \right)^k \right]. \tag{5}$$

From Eq. (5), rainfall quantile $X_T$ appears as a function of covariate $Z$. That is, Eq. (5) shows that various rainfall quantiles are calculated depending on the value of the covariate even at the same return level. Therefore, one of the problems to be solved in the non-stationary frequency analysis using a covariate is how to set the value of the covariate corresponding to a specific quantile. Since it is often required to have a single design rainfall depth in practice, it is very cumbersome to give a result of calculating rainfall quantiles of various values depending on a change in a covariate.

## 2.2 Metropolis-Hastings algorithm

The parameters of the GP distribution were estimated using the Metropolis-Hastings (MH) algorithm to account for uncertainty. This algorithm is one of the algorithms for the Markov Chain Monte Carlo (MCMC) sampling, which takes a sample from the posterior distribution of the parameter θ given the observation data Y. The MH algorithm starts with the initial parameter value $\theta_o$. Then, $N + M$ sequences of the parameter $\theta_i$ $(i = 1, \cdots, N + M)$ are generated through the following procedure:

1) The candidate parameter $\theta^*$ is generated from the proposal distribution $q(\theta^*|\theta_{i-1})$. At this time, the proposal distribution was applied to the truncated normal distribution with mean $\theta_{i-1}$ and variance $\Sigma$ in this study. The upper and lower limits of the truncated normal distirubtion corresponding to the upper and lower limits of the parameters were determined in advance.

2) Calculation of the reference value $T$ for adoption as follows:

$$T = \frac{\pi(Y|\theta^*)q(\theta_{i-1}|\theta^*)}{\pi(Y|\theta_{i-1})q(\theta^*|\theta_{i-1})}, \tag{6}$$

where $\pi(Y|\theta^*)$ and $\pi(Y|\theta_{i-1})$ are the likelihood values in the parameters $\theta^*$ and $\theta_{i-1}$, respectively, and are defined as follows:

$$\pi(Y|\theta) \sim \prod_{i=1}^{n} f(y_i), \tag{7}$$

where $f()$ is the probability density function of the GP distribution.

3) If $\min(1, T) > u$ is satisfied for a uniform random number $u$ between 0 and 1, $\theta_i =$

$\theta^*$, otherwise $\theta_i = \theta_{i-1}$.

The Markov chain, constructed through the initial $N$ iterations, converges to a chain that randomly samples parameters from the posterior distribution of parameters. At this time, the parameter sampled before the initial $N$ iterations should be discarded.

Before using the MH algorithm, it is necessary to determine the initial parameter $\theta_o$, the proposal distribution $q(\theta^*|\theta_{i-1})$, the initial iterative sampling number $N$, and the total iterative sampling number $N + M$. The choice of the initial parameter value $\theta_o$ is generally not sensitive to the results, while the choice of the proposal distribution $q(\theta^*|\theta_{i-1})$ is important. The general method is to use a normal distribution with mean $\theta_{i-1}$ and a constant covariance matrix $\Sigma$. It is recommended to select $\Sigma$ so that the adoption rate of $\min(1, T) >$ $u$ is 20 to 70 %. The number of iterations to be discarded, $N$, is known to be sufficient if more than 10 % of $M$ is applied, and the number of samples, $M$, should be secured enough to track the progress of the chain and converge the average values of the parameter posterior distribution.

The characteristics of the posterior distribution of parameters from the generated samples can be quantified. In general, the final estimated parameter $\bar{\theta}$ is calculated as follows:

$$\bar{\theta} = \frac{1}{M}\sum_{i=N+1}^{N+M} \theta_i. \tag{8}$$

In addition, the variance of the estimated parameters can be calculated from the generated samples.

## 3. Results

### 3.1 Selection of POT threshold

Since frequency analysis using POT excesses requires independent rainfall data greater than the threshold $x_o$, it is necessary to set $x_o$. One of the most commonly used methods of setting the appropriate $x_o$ is the mean residual life plot (Coles, 2001), and the results of applying it to the daily precipitation data at Busan and Seoul sites are shown in Figure 2. In general, a nonlinear curve appears in a section where a small $x_o$ is selected, and an

approximate straight line appears as $x_o$ increases. It is recommended to set $x_o$ in this straight section. From Figure 2, it can be found that the appropriate range of $x_o$ is in the range of 30 to 150 mm/day for both Busan and Seoul sites. In this study, $x_o$ = 50 mm / day was set as a threshold for the POT series in both Busan and Seoul sites. Mean residual life plots for all applied sites are shown in Figure S2 of Supplementary Material. In general, it can be recognized that it is feasible to set $x_o$ = 50 mm/day as the threshold for the POT time series at all sites.

[Figure 2. Mean residual life plot at (a) Busan and (b) Seoul sites. The solid line is the mean of the excesses of the threshold, and the dotted line is approximated 95 % confidence intervals.]

**3.2 Stationary frequency analysis**

The parameters of the GP distribution were estimated using the method of probability weighted moments (PWM) and MH algorithm, respectively. Although maximum likelihood estimation is an efficient method, it does not clearly show efficiency even in samples larger than 500 (Smith, 1985). The method of moments is generally known to be reliable except when the shape parameter is less than -0.2. When the likelihood that the shape parameter is less than 0 is high, PWM estimation is recommended (Hosking and Wallis, 1987). Figure 3 shows the result of PWM parameter estimation and the posterior distribution of parameters by the MH algorithm at Busan and Seoul sites. Since the MH algorithm does not return a single-valued parameter, but estimates the posterior distribution of the parameter, information about the uncertainty of the estimated parameter can be obtained. It can be recognized that the posterior distribution of the scale parameter converged to an appropriate range even though a relatively wide range of uniform distribution was assumed as the prior-distribution (the whole section of the horizontal axis in Figure 3). However, in the case of the shape parameter, it can be found that the uncertainty is formed relatively higher. That is, it can be seen that the uncertainty included when fitting the POT series of Busan and Seoul sites to the GP distribution is mainly due to the estimation of the shape parameter.

[Figure 3. Posterior distribution of parameters of stationary and non-stationary GP distribution. (a) Scale and (b) shape parameters at Busan site, and (c) scale and (d) shape parameters at Seoul site. The black vertical lines are a parameter calculated by PWM, which is expressed as a single value. The posterior distribution of parameters for the stationary GP distribution sampled using the MH algorithm is indicated by red lines. The posterior distribution of parameters for the non-stationary GP distribution is indicated by blue lines. The scale parameter of the non-stationary GP distribution using covariate is defined as a function of DPT. Therefore, the posterior distribution of the scale parameters were derived on the assumption that DPT was given at 20.2567 ℃ (Busan site) and 21.4958 ℃ (Seoul site), respectively.]

Table 1 shows the final estimated parameters at Busan and Seoul sites. The parameter estimation value of the MH algorithm was defined as the ensemble average of samples extracted by MCMC from the posterior distribution as mentioned in Eq. (8). The parentheses of the parameter estimation values by the MH algorithm in Table 1 are the coefficient of variation of the parameter. It can be found that the PWM and MH algorithms give similar parameter values for both scale and shape parameters. The negative logarithm likelihood (nllh) was also calculated similarly. From the above results, it can be recognized that estimating parameters by MH algorithm is applicable when attempting to fit the POT time series to the GP distribution, and information about the uncertainty of the estimated parameters is also obtainable. It can also be found that the coefficient of variation of the ensemble of scale parameters sampled by MCMC is less than 10 %, while the coefficient of variation of the ensemble of shape parameters is around 40 %. This means that the uncertainty of the shape parameters is relatively higher. Results for other sites tend to be similar to those obtained at Busan and Seoul sites. Results for other sites are shown in Table S1 of Supplemental Material.

[Table 1. Parameter estimation of stationary GP distribution at Busan and Seoul sites]

### 3.3 Non-stationary frequency analysis

For analyzing the non-stationary frequency of POT excesses, the non-stationary GP

distribution, in which the scale parameter was defined as a function of DPT or SAT on the day when the POT excesses occurred, was set as in Eq. (4). The parameters of the non-stationary GP distribution were estimated using the MH algorithm, and Figure 3 shows the posterior distribution of the parameters by the MH algorithm. Similar to the stationary GP distribution, the posterior distribution of the scale parameter converged to an appropriate range, although a relatively wide range of prior-distributions was assumed. However, it can be recognized that the uncertainty is still high in the case of the shape parameter.

The scale parameter finally estimated at Busan site using Eq. (8) is $\alpha = exp[2.2149 + 0.071078 \cdot Z]$ (where $Z$ is DPT), and the shape parameter is $k = -0.1123$. The coefficient of variation of the scale parameter was 7.66 % when the DPT was given at 20.2567 ℃, and the coefficient of variation of the shape parameter was 44.02 %. Therefore, when compared with the coefficient of variation of the parameters of the stationary GP distribution in Table 1, it can be recognized that the uncertainty in both the scale parameter and the shape parameter slightly decreased in the non-stationary GP distribution. However, in the scale parameter, these coefficients of variation were obtained under the assumption of a specific DPT, so if the range of the observed DPT was reflected, the coefficient of variation of the scale parameter of the non-stationary GP distribution would have a larger value. The AIC of the stationary model was AIC = 3264.84, and the AIC of the non-stationary model was calculated as AIC = 3247.61. From the viewpoint that the AIC of the non-stationary model is slightly smaller, it can be said that the non-stationary model has better performance in expressing the frequency of the POT excesses than the stationary model. The parameter estimation results of other sites also showed a similar trend to those of Busan site. In other words, under certain DPT or SAT conditions, the uncertainty of the scale and shape parameters of the non-stationary model was slightly reduced than that of the stationary model, and the AIC of the non-stationary model was calculated to be smaller than the AIC of the stationary model.

### 3.4 Uncertainty analysis

The final goal of frequency analysis is the estimation of rainfall quantiles, but the parameters of probability distribution required for estimation of quantiles as well as quantiles are inevitably uncertain since they are estimated from limited samples. Therefore, looking at

the uncertainty of the parameters of the probability distribution applied and the uncertainty of the quantile derived as a result of frequency analysis give important information to determine whether the model is applicable. In this study, the following dimensionless quantitation factors were defined to quantify the uncertainty between the stationary and non-stationary models:

$$m - factor = \frac{Width\ of\ 95\ PPU\ for\ parameter}{estimated\ parameter\ value}, \text{and} \tag{9}$$

$$h - factor = \frac{Width\ of\ 95\ PPU\ for\ rianfall\ quantile\ ensemble}{rainfall\ quantile\ estimate}, \tag{10}$$

where 95 PPU means 95 % predicted uncertainty of the corresponding variable (Abbaspour et al., 2007). In fact, m-factor and h-factor can be seen as quantification of confidence intervals of ensembles simulated by MCMC. That is, the m-factor and h-factor of the estimated value indicate how accurate the estimate is or how much uncertainty is (Ouarda et al., 2020). The greater the uncertainty of the parameter or rainfall quantile, the greater the value of 95 PPU. That is, the quantitation factors of uncertainty expressed by m-factor and h-factor reflect the diffusion or lack of precision of the ensemble sampled from the posterior distribution (Motavita et al., 2019).

A total of 6,000 parameter values were sampled from the posterior distribution of parameters for each of the stationary and non-stationary models, and 6,000 rainfall quantile ensemble corresponding to a return level of 100-year were generated. Eqs. (9) and (10) were used to quantify the uncertainty for parameters and the uncertainty for rainfall quantile. Table 2 shows the results at Busan and Seoul sites. For reference, the results of applying DPT or SAT as a covariate at other sites are shown in Table S2 of Supplementary Material. The parameters of the stationary GP distribution are $\alpha$ and $k$, whereas the parameters of the non-stationary GP distribution are $\alpha_1$, $\alpha_2$, and $k$, so for direct comparison, m-factor derived by converting $\alpha_1$ and $\alpha_2$ of the non-stationary GP distribution to $\alpha = exp[\alpha_1 + \alpha_2 DPT_r]$ were expressed together. Here, $DPT_r$ is a reference DPT, and 20.2567 °C for Busan site and 21.4958 °C for Seoul site, respectively. The reference DPT will be discussed in detail in the discussion section.

[Table 2. Uncertainty of stationary and non-stationary frequency analysis at Busan and Seoul sites]

The uncertainty of the parameters was first investigated for the m-factor of Eq. (9). It can be found that the uncertainty of the scale parameter of the non-stationary model is less than the uncertainty of the scale parameter of the stationary model under the condition given the reference DPT (10.9 % at Busan site, and 1.7 % at Seoul site). In the case of the shape parameter, Busan and Seoul sites showed different results. The uncertainty of the non-stationary model decreased at Busan site (10.2%), but increased at Seoul site (9.9%). This suggests that even if a non-stationary model is introduced, it is difficult to expect that the uncertainty resulting from parameter estimation of the GP distribution would be reduced. The fact that the uncertainty in the scale parameter has been shown to be reduced is the result from the condition under which a specific DPT is given, so it would be also difficult to argue that the uncertainty in the scale parameter has been reduced if changes in DPT are reflected.

The h-factor of rainfall quantile corresponding to the return level of 100-year was calculated in two ways. First, under the condition that the reference DPT is given (i.e., when the reference value of DPT is applied), the h-factor of the non-stationary model is reduced by 37 % (at Busan site) and 28 % (at Seoul site) than that of the stationary model. However, under the condition that all observed DPTs corresponding to POT excesses are applied, the uncertainty from parameter estimation and the effects from extreme values of the covariate are combined, and the h-factor of the non-stationary model exceeds the h-factor of the stationary model. That is, if samples of the scale parameter (i.e., $\alpha$) is made by combining all samples of the coefficients of the scale parameter (i.e., $\alpha_1$ and $\alpha_2$) and samples of all observed DPTs corresponding to each POT excess, the uncertainty of rainfall quantiles in the non-stationary model is greater than the uncertainty of rainfall quantiles in the stationary model. The amplification of the uncertainty in the non-stationary model is because, as can be seen from Eq. (4), samples of some extreme DPTs greatly scatter samples of the scale parameter of the non-stationary GP distribution. This can also be confirmed through the lower right figure of Figure 4(a) and (b). The width of the 95 PPU of the scale parameter of the non-stationary model corresponding to the value of the individual DPT is not significantly different from the width of the scale parameter of the stationary model. However, when all observed DPTs corresponding to the POT excesses are involved in sampling of the scale parameter, it can be recognized that the range of the 995 PPU of the scale parameter of the non-stationary model is

very wide.

[Figure 4. Changes in uncertainty for co-variate at (a) Busan and (b) Seoul sites. The upper left figures in Figure 4(a) and (b) show the POT series (blank line), and the ensemble average of stationary (blue line) and non-stationary (red line) rainfall quantile corresponding to the return level of 100-year. In the upper right figures, the ensemble average (blue line for stationary model, and red line for non-stationary model), and 95PPU of the stationary (blue dotted line) and non-stationary (red dotted line) rainfall quantile for the return level of 100-year are shown. The lower left figures show the h-factor of the stationary (blue line) and non-stationary (black line) rainfall quantile corresponding to the return level of 100-year. Red lines mean the average of black line. The lower right figures show the ensemble average (blue line for stationary model, and red line for non-stationary model), and 95 PPU of the stationary (blue dotted line) and non-stationary (red dotted line) scale parameter.]

The above results indicate that although the non-stationary model is better in fitting performance for the observed samples, it is difficult to admit that the non-stationary model is more reliable than the stationary model due to the influence of extreme values of the covariate when estimating rainfall quantile. Ouarda et al (2020) also produced similar results using the annual maximum rainfall series and the non-stationary GEV distribution.

We want to note here the condition in which the value of the covariate is given. In the upper left figure of Figure 4(a) and (b), the stationary quantile has a single value, while the ensemble average of the non-stationary quantile shows various values depending on the value of DPT. In addition, the 95 PPU of the stationary quantile has a constant range regardless of the value of the covariate, whereas the 95 PPU of the non-stationary quantile has a relatively wider range depending on the value of the covariate (see upper right figure in Figure 4(a) and (b)). This is due to the covariate dependence inherent in the scale parameter of the non-stationary GP distribution, as mentioned before. That is, since the range of the ensemble of the non-stationary rainfall quantile is a result of additionally reflecting the extreme values of the covariate in addition to the parameter uncertainty, it is more likely to be formed relatively wider than the range of the ensemble of the stationary rainfall quantile. It should be noted, however, that the

width of the non-stationary 95 PPU for a particular covariate value is less than the width of the stationary 95 PPU.

In fact, since the covariate corresponding to each POT excess is a known value, the h-factor of the rainfall quantile corresponding to each POT excess can be obtained (see lower left figure in Figure 4(a) and (b)). Given the value of covariate, it can be recognized that the non-stationary h-factor is smaller than the stationary h-factor. That is, if the value of the covariate of the non-stationary model can be determined, there is a room to say that the non-stationary frequency analysis is better in terms of reliability than the stationary frequency analysis.

Figure 5 shows the empirical distribution of rainfall quantile corresponding to the return level of 100-year using DPT observed at Busan and Seoul sites. Note that the non-stationary GP distribution using the covariate returns rainfall quantile of various values depending on the DPT corresponding to the POT excess. As can be seen in Figure 5, the non-stationary frequency analysis can provide an empirical distribution of rainfall quantile in the present condition of DPT and in the future condition of elevated DPT due to global warming. Therefore, the change in rainfall quantile considering global warming can be expressed explicitly. While rainfall extremes derived from climate models have significant bias and uncertainty, relatively reliable climate model outputs can be obtained for DPT (O'Gorman, 2012; Lenderink and Attema, 2015; Farnham et al., 2018). Therefore, it can be said that the non-stationary frequency analysis using DPT or SAT has an advantageous structure for examining the effect of global warming on rainfall quantile (Wasko and Sharma, 2017; Lee et al., 2020).

[Figure 5. Rainfall quantile estimates at (a) Busan and (b) Seoul sites for return level of 100-year using observed dew-point temperature and global warming scenarios. The stationary rainfall quantile is indicated as a blue vertical line since it is a single value. The non-stationary rainfall quantiles were calculated using the average of the parameter ensemble sampled by MCMC and the DPT observed on the day of POT excesses (red dotted line). In this figures, 'NS (3 ℃ up)' is an empirical distribution of rainfall quantile derived using DPTs that add 3 ℃ to DPTs observed on the day of POT excesses. Likewise, 'NS (5 ℃ up)' is an empirical distribution of rainfall quantile under the scenario condition where DPT has risen 5 ℃ due to global warming.]

## 4. Discussion

### 4.1 Reference covariate

As described above, when performing the uncertainty analysis of the non-stationary frequency analysis, an undesired disturbance in which the ensemble of rainfall quantile is excessively dispersed due to some extreme covariate values appears. Since the value of the covariate is the data observed on the day that the POT excess occurred (i.e., a deterministic variable), analyzing the uncertainty in rainfall quantile by randomly sampling the value of DPT or SAT from a predefined probability distribution of covariate is likely to result in overestimating uncertainty. We thought that the uncertainty analysis of randomly sampling the values of covariate from a predefined distribution of covariate was not feasible. The method of randomly sampling the value of the covariate in this study was implemented under the condition that all observed covariate samples corresponding to POT excesses were applied. Therefore, this study investigated the relationship between the value of covariate and rainfall quantile.

From Eq. (5), the DPT value (i.e., reference DPT) of the non-stationary GP distribution that returns the rainfall quantile equal to the stationary GP distribution can be calculated (reference SAT can be calculated in the same way). Figure 6 shows an example of determining a reference DPT. The results of calculating the reference DPT at Busan and Seoul sites indicate that the reference DPT increases as the return level increases. The right figure in Figure 6(a) and (b) shows the histogram of DPT corresponding to POT excesses. The distribution of DPT is slightly distorted to the left. It can be found that the reference DPT corresponding to various return levels at Busan and Seoul sites is similar to the location of the mode of the DPT distribution. This fact reveals that covariate values that deviate significantly from the reference covariate (i.e., some extreme values of the covariate) amplify the uncertainty of rainfall quantile from the non-stationary frequency analysis. From the results of regression analysis of rainfall quantile for various return levels and the corresponding reference DPT, the relationship of *DPT = 18.8589RL^0.01555* (where *RL* is the return level in year and the unit of *DPT* is °C) was obtained at Busan site. At Seoul site, a relationship of *DPT = 19.8540RL^0.01728* was obtained. The coefficient of determination of the regression analysis was 0.99 or higher at

Busan and Seoul sites. From these results, the reference DPT corresponding to the return level of 100-year at Busan site could be applied to 20.2567 °C and Seoul site to 21.4958 °C. As shown in Figure 6 and Figures S3 and S4 of Supplementary Material, the value of the reference covariate is almost completely dependent on the return level. It should be noted that the return level and the reference covariate are proportional to each other at some sites, and are inversely proportional to other sites. This means that it is not easy to identify a single covariate value corresponding to a rainfall quantile. In this study, we tried to overcome the problem of random sampling of covariates by introducing the concept of reference covariate when estimating rainfall quantile and analyzing its uncertainty from non-stationary frequency analysis based on covariate. From a practical point of view, how to set the value of the reference covariate may be an important research topic in the covariate-based non-stationary frequency analysis.

[Figure 6. Selection of reference dew-point temperature for estimating rainfall quantiles at (a) Busan and (b) Seoul sites. In this figure, 'RF' refers to the empirical relative frequency of DPT on the day of POT excess.]

Figure 7(a) shows the values of the negative log likelihood function of the stationary model and the non-stationary models at 13 sites. The stationary model, the SAT-based non-stationary model, and the DAT-based non-stationary model were found to have no significant difference in the fit performance with the observed POT excesses. Figure 7(b) shows the h-factor of rainfall quantile corresponding to the return level of 100-year. When all the values of covariate observed on the day of POT excesses are considered ("DPT" and "SAT" in Figure 7(b)), at all sites except Mokpo site, the non-stationary h-factor is greater than the stationary h-factor. However, when the reference covariate is applied, the non-stationary h-factor is smaller than the stationary h-factor. Results from 13 sites and most of the non-stationary models using SAT or DPT as a covariate indicate that how to determine the appropriate value of the covariate corresponding to the rainfall quantile plays an important role in securing the reliability of the non-stationary frequency analysis.

[Figure 7. Performance of stationary and non-stationary frequency analysis models. At Site ID, 1: Ghangreung, 2: Seoul, 3: Incheon, 4: Chupungryeong, 5: Pohang, 6: Daegu, 7: Jeonju, 8: Ulsan, 9: Ghwangju, 10: Busan, 11: Mokpo, 12: Yeosu and 13: Jeju site.]

The uncertainty of the non-stationary frequency analysis for various sample size changes was analyzed using a reference DPT corresponding to the return level of 100-year. In the first case, POT and covariate series of the last 10 years from 2008 to 2017 were applied, and frequency analysis was performed by extending the data period in the past direction for 5 years. Figure 8 shows the uncertainty of rainfall quantile under the condition that the reference DPT is given. Generally, the h-factor of the non-stationary frequency analysis is smaller than the stationary frequency analysis. It can be found that for the h-factor to be less than 0.5, the non-stationary frequency analysis requires a data period of about 40 years (at Busan site) and 75 years (at Seoul site), while the stationary frequency analysis requires more than 100 years. The data period required to achieve a certain level of h-factor can play an important role in optimal model selection. As in other regions, the observed data period in Korea varies widely from site to site. If the data period is short and there is no significant difference in performance (both in terms of goodness of fit and uncertainty) between the stationary model and the non-stationary model, it can be said that it is better to apply a stationary model with a relatively well-established methodology. However, in terms of uncertainty, if the value of the reference covariate can be well defined, the results in Figure 8 show that the non-stationary model can estimate rainfall quantile with the same level of uncertainty even with relatively shorter data periods. That is, when frequency analysis is performed using samples of the same data period at a site, if the appropriate covariate is applied and the reference value of the covariate is appropriately determined, it can be said that the rainfall quantile estimated from the non-stationary model is more reliable than the rainfall quantile estimated from the stationary model.

[Figure 8. Effect of the number of samples on the uncertainty of rainfall quantile using reference dew-point temperature.]

## 4.2 Uncertainty of rate of change

Through Figure 5, we have seen that the non-stationary frequency analysis using DPT has an advantageous structure for examining the effect of global warming on rainfall quantile. In this section, we extend the concept of Figure 5 a little further to investigate the uncertainty about the rate of change in rainfall quantile for global warming. Here, the rate of change is defined as [future rainfall quantile - present rainfall quantile] / [present rainfall quantile]. That is, a rate of change of 0.2 means that the future rainfall quantile will increase by 20 % from the present rainfall quantile. In most global warming scenarios, the state of DPT increases, so the case where the change rate is less than 0 is not considered in this study. In fact, it is not difficult to consider.

Let us assume that rainfall quantile for the return level of T-year in the present DPT state is $X_p^T$, and rainfall quantile in the future DPT state is $X_f^T$. At this time, $X_p^T$ and $X_f^T$ are composed of ensemble sampled by MCMC under the conditions given the present and future reference DPT, respectively. The probability that the rainfall quantile $X_f^T$ in the future DPT state increases more than $\alpha \times 100$ (%) than the rainfall quantile $X_p^T$ in the present DPT condition, that is, the probability $P_\alpha^T$ that the rate of change becomes more than $\alpha$ can be defined as follows:

$$P_\alpha^T = P\left[X_f^T \geq (1+\alpha)X_p^T\right] = 1 - \int_0^\infty F_f^T\left[(1+\alpha)X_p^T\right] \cdot f_p^T\left[X_p^T\right] dX_p^T, \qquad (11)$$

where $F_f^T[]$ is the cumulative probability distribution function of $X_f^T$ in the future DPT state, and will behave depending on the DPT rise in the future global warming scenario. The probability distribution of $X_p^T$ in the present DPT state was expressed as $f_p^T[]$. From Eq. (11), it can be recognized that $P_\alpha^T$ increases as the future DPT increase increases, and decreases as the rate of change increases.

When frequency analysis using the Bayesian approach is performed, a large number of samples for $X_p^T$ and $X_f^T$ can be obtained through MCMC simulation, so instead of calculating $P_\alpha^T$ using Eq. (11), it is possible to numerically calculate $P_\alpha^T$ from the generated samples. Figure 9 shows the probability that the rainfall quantile for the return level of 100-year will exceed a certain rate of change under various conditions (ΔDPT) in a global warming scenario expressed as a rise in DPT. That is, the probability that the rainfall quantile for the

return level of 100-year increases by 20 % or more in a scenario condition in which the state of DPT increases by 6 ℃ at Busan site is about 80 %.

[Figure 9. Likelihood of increase over change rate of rainfall quantile for return level of 100-year.]

When Figure 9 is substituted for a specific DPT rise scenario, the reliability of the rate of change in rainfall quantile can be obtained as explained below. Figure 10 describes the procedure for analyzing the rate of change in rainfall quantile for the return level of 100-year under the DPT 4 ℃ rise scenario. The upper left figures in Figure 10(a) and (b) show the probability distribution of rainfall quantile ensemble at Busan and Seoul sites, respectively. One can see that the probability distribution of $X_f^T$ is shifted to the right. Using the information on these probability distributions and the concept of Eq. (11), the likelihood of increase over change rate (LoI), $P_\alpha^T$, can be drawn (see the upper right subplot). Since LoI is the probability that the rate of change of rainfall quantile for a specific return level is greater than or equal to $\alpha$ in a specific DPT rising condition, the probability that the rate of change is less than or equal to $\alpha$ is $1 - P_\alpha^T$. That is, the cumulative probability distribution of rate of change becomes $1 - P_\alpha^T$, which is shown in the lower right. The probability distribution of rate of change can be obtained numerically from the cumulative probability distribution of rate of change, and it is shown in the lower left. The ensemble average of the rate of change of rainfall quantile for the return level of 100-year at Busan site was 0.3138 (0.3742 at Seoul site) and the standard deviation of the ensemble was 0.2734 (0.3298 at Seoul site).

[Figure 10. Procedure for analyzing uncertainty in rate of change. In upper left figures, the blue line is the probability distribution of $X_p^T$ in the present condition, and the red line is the probability distribution of $X_f^T$ in the DPT 4 ℃ rising condition. In the lower left figures, the section of the standard deviation was colored in pink.]

The uncertainty of the parameters estimated in the frequency analysis will influence the estimation of the rate of change in future climate change scenarios. An ensemble of rainfall quantile can be obtained from various parameter combinations sampled by MCMC, and an ensemble of future rainfall quantile can also be obtained by applying climate change scenarios to covariates. A simple comparison of the ensemble average of rainfall quantile derived from present and future DPT states using a reference DPT makes it possible to obtain an average rate of change, but it is impossible to determine how reliable the rate of change is. Through the procedure presented in Figure 10, one can recognize that it is possible to quantify the uncertainty inherent in the rate of change. It should be noted, however, that this uncertainty analysis of rate of change only considered the uncertainty that comes from parameter estimation. When analyzing the uncertainty of the rate of change, the uncertainty arising from the selection of probability distributions for frequency analysis and the uncertainty resulting from the choice of covariates should also be addressed. In addition, the uncertainty arising from various climate change scenarios should be treated as important.

## 5. Conclusion

In this study, stationary and non-stationary frequency analysis was performed using daily precipitation data from 13 major sites in Korea. Daily precipitation data for frequency analysis was extracted based on the POT approach. As a threshold for extracting the POT series, it was confirmed that a value between 30 and 150 mm/day was appropriate from the results of plotting the Mean residual life plot. Both Busan and Seoul site have finally set 50 mm/day as the threshold of the POT excesses. The POT series was adapted to the GP distribution, and as a result of estimating the parameters using the PWM and MH algorithms, it was confirmed that the parameter estimation of the GP distribution by the MH algorithm is applicable. Confirmation of applicability to the MH algorithm means that information on the empirical probability distribution of the estimated GP distribution parameters can be obtained.

The non-stationarity of the POT series was implemented by expressing the scale parameter of the GP distribution as a function of the DPT or SAT observed on the day of the POT excess. The AIC of the non-stationary GP distribution using the covariate was calculated to be slightly smaller than the AIC of the stationary GP distribution. However, since the difference was

thought to be likely to change in any way during the parameter estimation process, it was recognized that the performance in terms of data fitness of the stationary and non-stationary GP distributions was almost similar. On the other hand, since the non-stationary frequency analysis using covariate can separately provide the empirical distribution of rainfall quantile at the current covariate level and the empirical distribution of rainfall quantile at the covariate level changed due to global warming, changes in rainfall quantile considering climate change can be expressed explicitly.

In this study, rainfall quantile for various parameter combinations was simulated using MCMC sampling from the posterior distribution of parameters derived by the MH algorithm. Under the condition considering all observed ranges of covariate, it was found that the uncertainty of the non-stationary model was calculated to be greater than the uncertainty of the stationary model since the effects of extreme values of covariate were added to the uncertainty resulting from parameter estimation. In other words, although the performance in terms of goodness of fit was better for the non-stationary model, it was difficult to say that the results of the non-stationary model were more reliable than the results of the stationary model because of the non-stationarity from the variation of the covariate when estimating rainfall quantile. However, in this study, the concept of reference covariate was introduced to prevent excessive dispersion of rainfall quantile ensemble due to extreme values of covariate. That is, it was suggested that the reliability of the non-stationary frequency analysis could be superior to the reliability of the stationary frequency analysis under the condition that an appropriate reference covariate is given. For reference, it was found that it was necessary to change the reference covariate in response to the return level of the rainfall quantile.

The focus of this study was on how to examine the relative superiority of the stationary and non-stationary models when performing frequency analysis. When considering the uncertainty of the parameter of probability distribution, which is mainly caused by the limited sample size, it was thought be insufficient to evaluate the relative goodness of the stationary and non-stationary models only by evaluating the fitness of the sample using the estimated parameter. This study was promoted from the viewpoint that a model with smaller uncertainty inherent in rainfall quantile, which is the result of frequency analysis, was better. From this point of view, it was found that the covariate-based non-stationary frequency analysis could be a better model than the stationary frequency analysis if the reference covariate was properly

given. In addition, it was recognized that the uncertainty of the rate of change of rainfall quantile in future covariate conditions could also be identified by using the rainfall quantile ensemble in present and future covariate conditions that can be obtained in the uncertainty analysis process.

## Acknowledgments

This work was supported by the National Research Foundation of Korea (NRF) grant funded by the Korea government (MSIT) (No. NRF-2019R1A2C1003114). The authors also this work was supported by Korea Environment Industry & Technology Institute(KEITI) through Smart Water City Research Program, funded by Korea Ministry of Environment(MOE)(2019002950004)

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

**Table captions:**

**Table 1. Parameter estimation of stationary GP distribution at Busan and Seoul sites**

**Table 2. Uncertainty of stationary and DPT-based non-stationary frequency analysis at Busan and Seoul sites**

Table 1. Parameter estimation of stationary GP distribution at Busan and Seoul sites

| Site | Parameter | PWM | MH |
|------|-----------|------|------|
| Busan | α | 33.5972 | 33.966 (8.54 %) |
| | k | -0.1423 | -0.1477 (47.44 %) |
| | nllh | 1630.38 | 1630.42 |
| Seoul | α | 34.9666 | 35.1785 (8.93 %) |
| | k | -0.1633 | -0.1772 (38.59 %) |
| | nllh | 1340.82 | 1340.87 |

Table 2. Uncertainty of stationary and DPT-based non-stationary frequency analysis at Busan and Seoul sites

| Site | Factor | Parameter | Stationary | Non-stationary |
|------|--------|-----------|------------|----------------|
| Busan | $m-$ factor | $\alpha_1$ | | 0.5463 |
| | | $\alpha_2$ | | 0.8700 |
| | | α | 0.3278 | 0.2920 |
| | | k | 1.7507 | 1.5717 |
| | $h-$ factor | 100-yr | 0.7595 | 0.4771 (1.0274) |
| Seoul | $m-$ factor | $\alpha_1$ | | 0.7127 |
| | | $\alpha_2$ | | 0.8588 |
| | | α | 0.3407 | 0.3349 |
| | | k | 1.4204 | 1.5613 |
| | $h-$ factor | 100-yr | 0.7421 | 0.5331 (1.0273) |

**Figure captions:**

Figure 1. Sensitivity of 95 % daily rainfall depth to dew-point temperature at (a) Busan and (b) Seoul sites.

Figure 2. Mean residual life plot at (a) Busan and (b) Seoul sites. The solid line is the mean of the excesses of the threshold, and the dotted line is approximated 95% confidence intervals.

Figure 3. Posterior distribution of parameters of stationary and non-stationary GP distribution. (a) Scale and (b) shape parameters at Busan site, and (c) scale and (d) shape parameters at Seoul site. The black vertical lines are a parameter calculated by PWM, which is expressed as a single value. The posterior distribution of parameters for the stationary GP distribution sampled using the MH algorithm is indicated by red lines. The posterior distribution of parameters for the non-stationary GP distribution is indicated by blue lines. The scale parameter of the non-stationary GP distribution using covariate is defined as a function of DPT. Therefore, the posterior distribution of the scale parameters were derived on the assumption that DPT was given at 20.2567 °C (Busan site) and 21.4958 °C (Seoul site), respectively.

Figure 4. Changes in uncertainty for co-variate at (a) Busan and (b) Seoul sites. The upper left figures in Figure 4(a) and (b) show the POT series (blank line), and the ensemble average of stationary (blue line) and non-stationary (red line) rainfall quantile corresponding to the return level of 100-year. In the upper right figures, the ensemble average (blue line for stationary model, and red line for non-stationary model), and 95PPU of the stationary (blue dotted line) and non-stationary (red dotted line) rainfall quantile for the return level of 100-year are shown. The lower left figures show the h-factor of the stationary (blue line) and non-stationary (black line) rainfall quantile corresponding to the return level of 100-year. Red lines mean the average of black line. The lower right figures show the ensemble average (blue line for stationary model, and red line for non-stationary model), and 95 PPU of the stationary (blue dotted line) and non-stationary (red dotted line) scale parameter.

**Figure 5. Rainfall quantile estimates at (a) Busan, and (b) Seoul sites for return level of 100-year using observed dew-point temperature and global warming scenarios. The stationary rainfall quantile is indicated as a blue vertical line since it is a single value. The non-stationary rainfall quantiles were calculated using the average of the parameter ensemble sampled by MCMC and the DPT observed on the day of POT excesses (red dotted line). In this figures, 'NS (3 °C up)' is an empirical distribution of rainfall quantile derived using DPTs that add 3 °C to DPTs observed on the day of POT excesses. Likewise, 'NS (5 °C up)' is an empirical distribution of rainfall quantile under the scenario condition where DPT has risen 5 °C due to global warming.**

**Figure 6. Selection of reference dew-point temperature for estimating rainfall quantiles at (a) Busan and (b) Seoul sites. In this figure, 'RF' refers to the empirical relative frequency of DPT on the day of POT excess.**

**Figure 7. Performance of stationary and non-stationary frequency analysis models. At Site ID, 1: Ghangreung, 2: Seoul, 3: Incheon, 4: Chupungryeong, 5: Pohang, 6: Daegu, 7: Jeonju, 8: Ulsan, 9: Ghwangju, 10: Busan, 11: Mokpo, 12: Yeosu and 13: Jeju site.**

**Figure 8. Effect of the number of samples on the uncertainty of rainfall quantile using reference dew-point temperature.**

**Figure 9. Likelihood of increase over change rate of rainfall quantile for return level of 100-year.**

**Figure 10. Procedure for analyzing uncertainty in rate of change. In upper left figures, the blue line is the probability distribution of $X_p^T$ in the present condition, and the red line is the probability distribution of $X_f^T$ in the DPT 4 °C rising condition. In the lower left figures, the section of the standard deviation was colored in pink.**

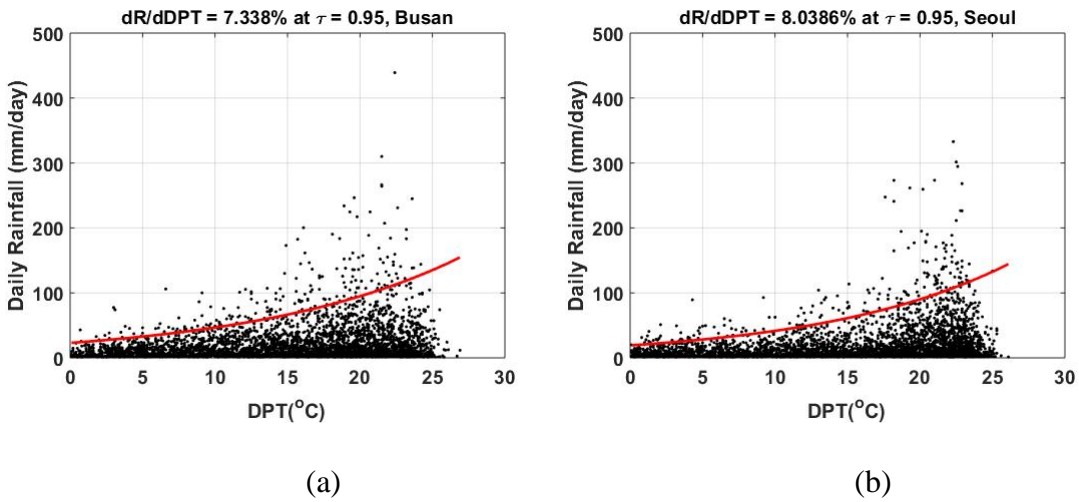

(a)                                                        (b)

Figure 1. Sensitivity of 95 % daily rainfall depth to dew-point temperature at (a) Busan and (b) Seoul sites.

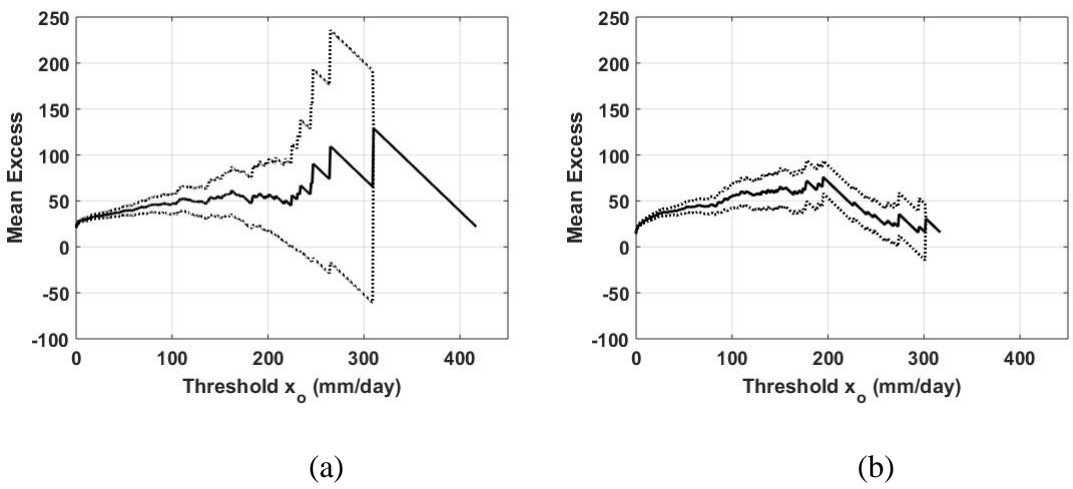

(a)                                                        (b)

Figure 2. Mean residual life plot at (a) Busan and (b) Seoul sites. The solid line is the mean of the excesses of the threshold, and the dotted line is approximated 95% confidence intervals.

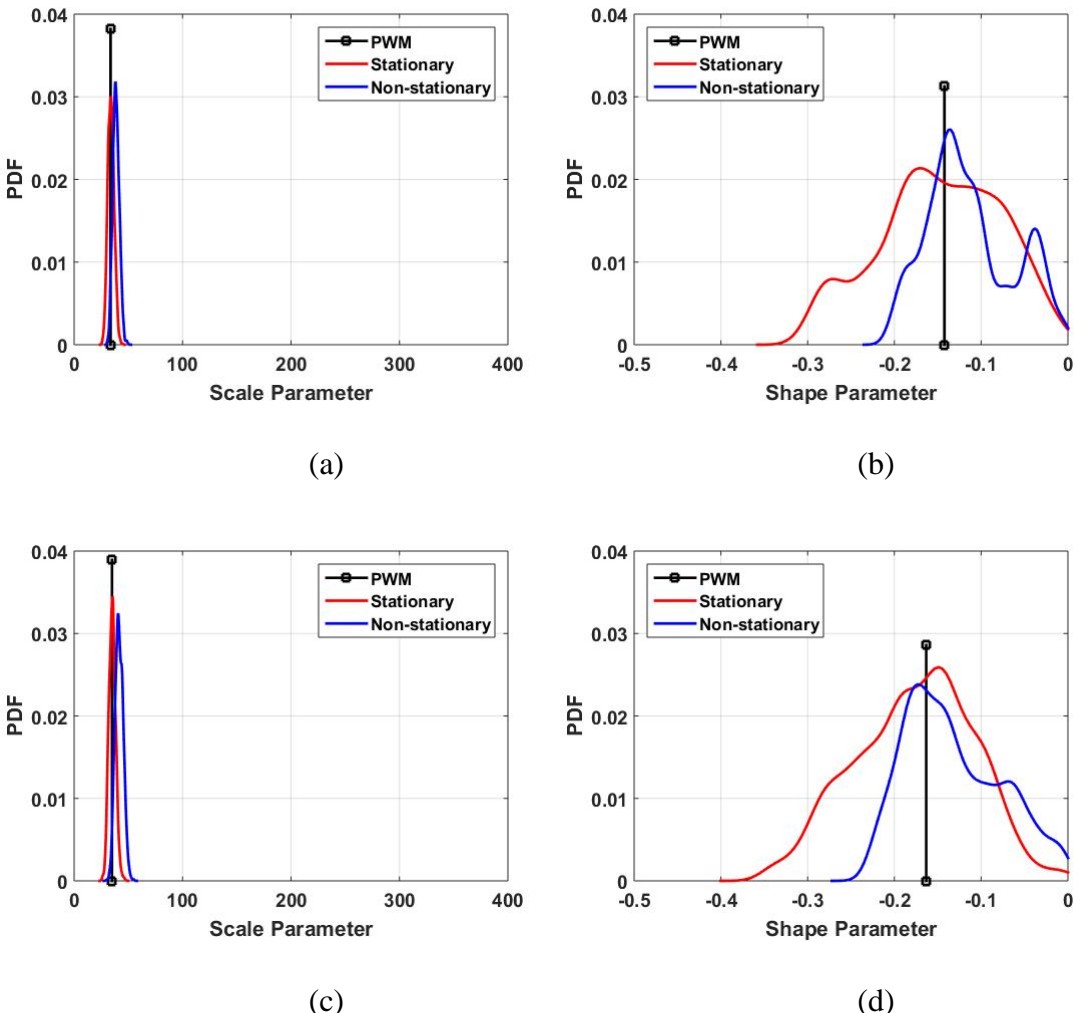

Figure 3. Posterior distribution of parameters of stationary and non-stationary GP distribution. (a) Scale and (b) shape parameters at Busan site, and (c) scale and (d) shape parameters at Seoul site. The black vertical lines are a parameter calculated by PWM, which is expressed as a single value. The posterior distribution of parameters for the stationary GP distribution sampled using the MH algorithm is indicated by red lines. The posterior distribution of parameters for the non-stationary GP distribution is indicated by blue lines. The scale parameter of the non-stationary GP distribution using covariate is defined as a function of DPT. Therefore, the posterior distribution of the scale parameters were derived on the assumption that DPT was given at 20.2567 ℃ (Busan site) and 21.4958 ℃ (Seoul site), respectively.

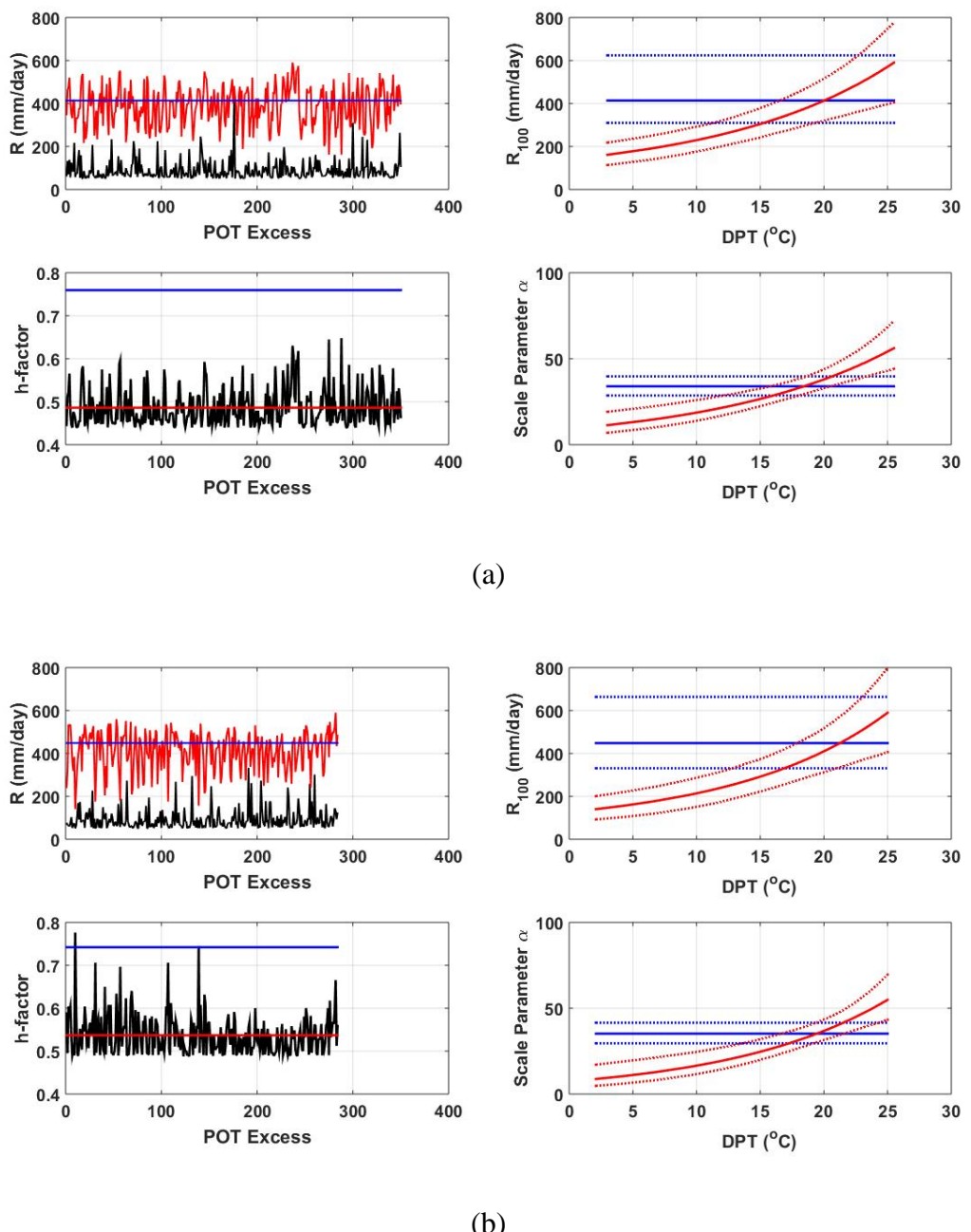

(a)

(b)

Figure 4. Changes in uncertainty for co-variate at (a) Busan and (b) Seoul sites. The upper left figures in Figure 4(a) and (b) show the POT series (blank line), and the ensemble average of stationary (blue line) and non-stationary (red line) rainfall quantile corresponding to the return level of 100-year. In the upper right figures, the ensemble average (blue line for stationary model, and red line for non-stationary model), and 95PPU of the stationary (blue dotted line) and non-stationary (red dotted line) rainfall quantile for the return level of 100-year are shown. The lower left figures show the h-factor of the stationary (blue line) and non-stationary (black

line) rainfall quantile corresponding to the return level of 100-year. Red lines mean the average of black line. The lower right figures show the ensemble average (blue line for stationary model, and red line for non-stationary model), and 95 PPU of the stationary (blue dotted line) and non-stationary (red dotted line) scale parameter.

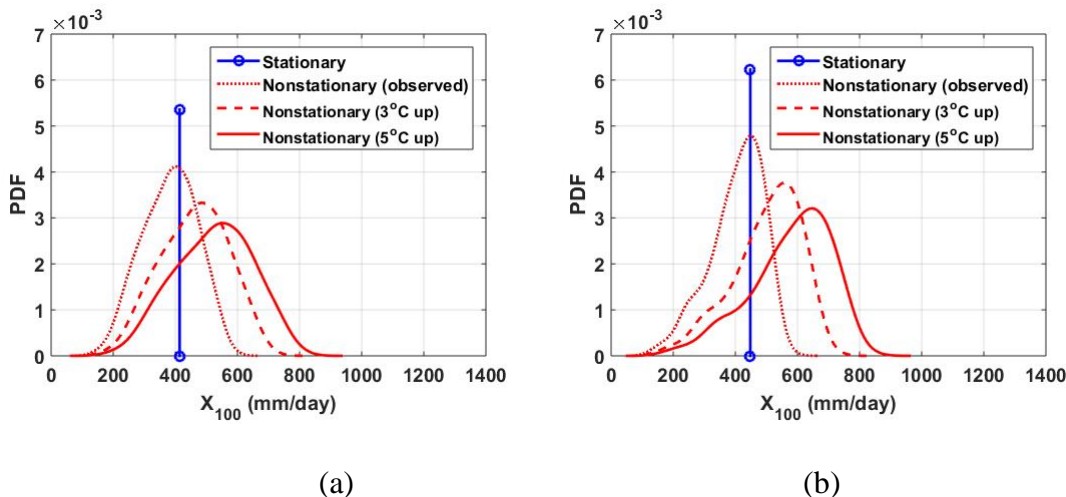

(a)                                                  (b)

Figure 5. Rainfall quantile estimates at (a) Busan, and (b) Seoul sites for return level of 100-year using observed dew-point temperature and global warming scenarios. The stationary rainfall quantile is indicated as a blue vertical line since it is a single value. The non-stationary rainfall quantiles were calculated using the average of the parameter ensemble sampled by MCMC and the DPT observed on the day of POT excesses (red dotted line). In this figures, 'NS (3 ℃ up)' is an empirical distribution of rainfall quantile derived using DPTs that add 3 ℃ to DPTs observed on the day of POT excesses. Likewise, 'NS (5 ℃ up)' is an empirical distribution of rainfall quantile under the scenario condition where DPT has risen 5 ℃ due to global warming.

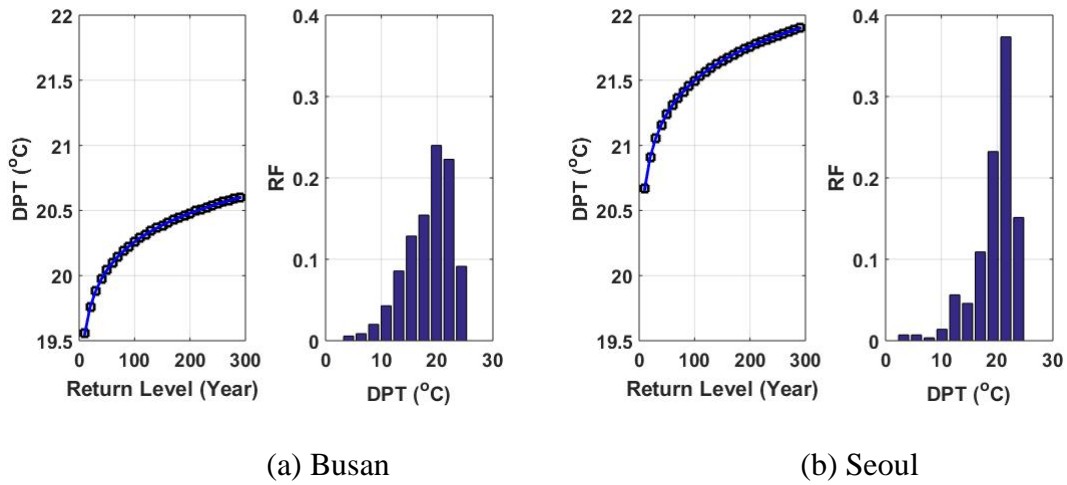

(a) Busan                  (b) Seoul

Figure 6. Selection of reference dew-point temperature for estimating rainfall quantiles at (a) Busan and (b) Seoul sites. In this figure, 'RF' refers to the empirical relative frequency of DPT on the day of POT excess.

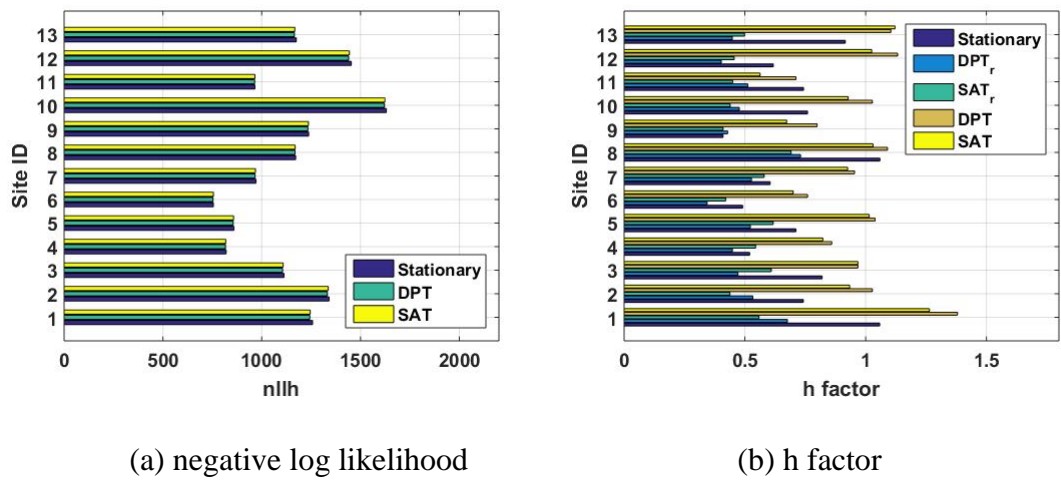

(a) negative log likelihood            (b) h factor

Figure 7. Performance of stationary and non-stationary frequency analysis models. At Site ID, 1: Ghangreung, 2: Seoul, 3: Incheon, 4: Chupungryeong, 5: Pohang, 6: Daegu, 7: Jeonju, 8: Ulsan, 9: Ghwangju, 10: Busan, 11: Mokpo, 12: Yeosu and 13: Jeju site.

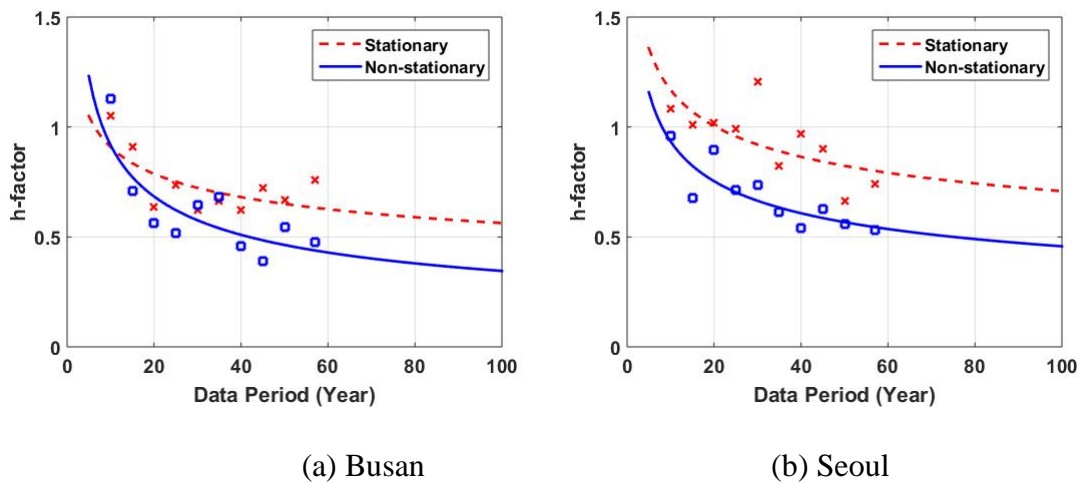

(a) Busan    (b) Seoul

Figure 8. Effect of the number of samples on the uncertainty of rainfall quantile using reference dew-point temperature.

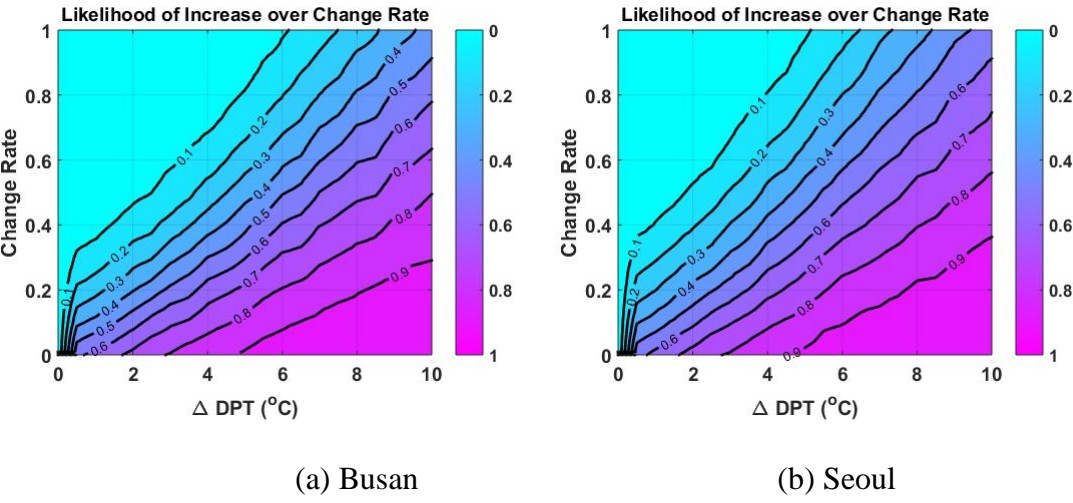

(a) Busan    (b) Seoul

Figure 9. Likelihood of increase over change rate of rainfall quantile for return level of 100-year.

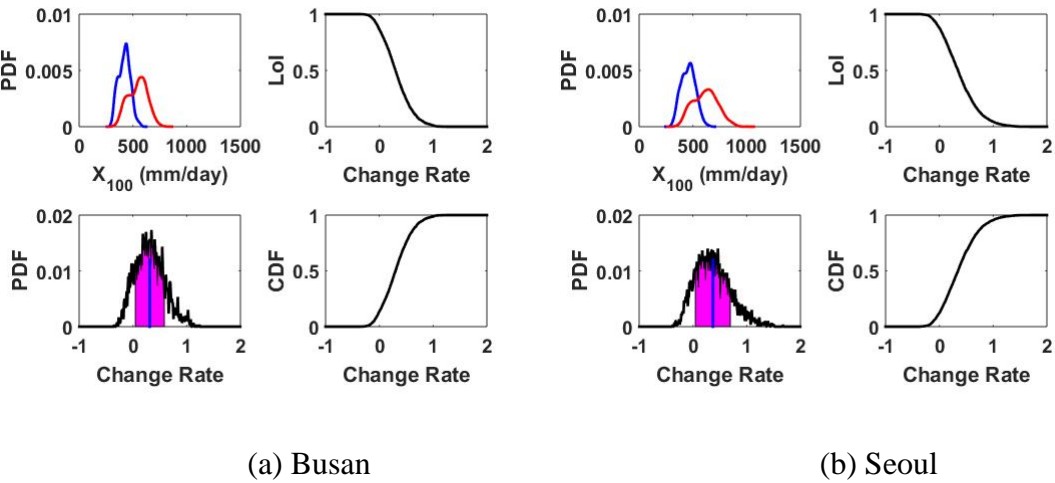

(a) Busan             (b) Seoul

Figure 10. Procedure for analyzing uncertainty in rate of change. In upper left figures, the blue line is the probability distribution of $X_p^T$ in the present condition, and the red line is the probability distribution of $X_f^T$ in the DPT 4 ℃ rising condition. In the lower left figures, the section of the standard deviation was colored in pink.