# Peer review of "Uncertainty in non-stationary frequency analysis of Korea's daily rainfall POT excesses associated with covariates"

_Hydrology and Earth System Sciences, 2020_

## Referee Comment (RC1) · Anonymous Referee #1 · 25 May 2020

The manuscript is overall well-written but needs to be checked for English thoroughly to increase readability. The reference list is appropriate and includes recent works on the topic. However, some improvements in the structure of the manuscript are needed, since some descriptions of the methodology are found in the discussion section. About the content of the manuscript, the analysis rely on just 2 stations in Korea, and only considers one covariate (daily dew point temperature), consequently the whole reads like a technical report rather than a scientific contribution to HESS. The interesting aspect is the model evaluation that is not based on model fit but on the uncertainties

on rainfall quantiles using a Bayesian framework. Overall the "reference DPT" concept used in the paper is interesting but not well defined and deserves a better explanation in the method section and discussion since it has practical interest for non-stationary analyses.

I would recommend adding more stations (obviously there are enough stations in Korea for such an analysis = https://doi.org/10.1002/joc.2068) and more importantly compare different covariates commonly used in non-stationary frequency analysis of extreme rainfall in the context of Korea (see my comment below about the lack of description on how the covariate is selected and used). These two recommendations would increase the representativeness of the results but also provide regional insights for Korea. At the current state of the manuscript, the reader cannot know if these results are only valid for these two stations and with this covariate.

Some specific comments:

My first comment is about the title, quite long and not very informative of the main scientific results of the paper. Something like this title might be better: Uncertainty of non-stationary frequency analysis applied to extreme rainfall in Korea.

Abstract, line 28, this sentence it not clear, maybe too general : "However, since the parameters of the estimated probability distribution contain a lot of uncertainty"

Abstract, line 40-42, this whole section below is quite trivial. Of course when a wrong covariate is selected in the POT model there is a stronger uncertainty. . . I don't see a major finding here.

Overall the abstract needs a major upgrading to better present the main findings of the study

Page 3, line 74 : change "in many documents" by "by many authors"

Page 3, lines 99-102: these sentences are not very clear.

[Figure]

Page 4, line 111, Why the authors are considering dew point temperature as a covariate for daily rainfall extremes for their stations in Korea ? Do previous works justify this choice ? No justification is given.

Page 5: line 130, "Daily rainfall depth of 0.1 mm or more was applied to the analysis," is not very clear. Does this mean you consider daily rainfall lower to 0.1 mm =0 ? Is this related to rain gauges uncertainties ? How the results can be impacted by this choice ?

Page 7, line 170: I guess you mean here instead the "scale" parameter

There is no indication on how the covariate is included in the model; is it the Dew point temperature the same day of the extreme rainfall event? at the starting day of a rainfall event or its peak ? On the opposite, is it computed for the week, or the months before the event ? No information is provided here.

Page 12, line 327, "However, under the condition that DPT is not given in advance" not clear to me. Do you mean you draw randomly DPT values instead of the values corresponding to the days with extreme rainfall?

Page 13: the whole page/paragraph is quite long and not very clear to me, it could be shortened to the main findings.

Page 14, line 366, this equation should be in the Method section

The beginning of section 4.1 is obviously not a discussion and should be in the results section

Page 14, lines 386-390, this information should be in the method section, we should not discover in the discussion section how the covariate was implemented in the model. . . see my comment above.

The issue of setting a reference covariate to a given return level is an interesting aspect. I believe it is necessary to first analyze the response of extreme rainfall to different

values taken by the covariate, and as mentioned here it is difficult to identify a single value of the covariate related to a high risk of extreme rainfall. Yet this aspect needs more discussion, I don't see an added value of randomly selecting a covariate from a pre-defined distribution (of the covariate).

Page 16, line 423; is this not totally expected?

Conclusions, lines 498-501, these sentences are elements of context and should not appear here but rather in the introduction.

Table 2: units?

---

## Author Comment (AC1) · 22 Jun 2020

Comment #1 :

The manuscript is overall well-written but needs to be checked for English thoroughly to increase readability. The reference list is appropriate and includes recent works on the topic. However, some improvements in the structure of the manuscript are needed, since some descriptions of the methodology are found in the discussion section. About the content of the manuscript, the analysis rely on just 2 stations in Korea, and only con-

siders one covariate (daily dew point temperature), consequently the whole reads like a technical report rather than a scientific contribution to HESS. The interesting aspect is the model evaluation that is not based on model fit but on the uncertainties on rainfall quantiles using a Bayesian framework. Overall the "reference DPT" concept used in the paper is interesting but not well defined and deserves a better explanation in the method section and discussion since it has practical interest for non-stationary analyses. I would recommend adding more stations (obviously there are enough stations in Korea for such an analysis = https://doi.org/10.1002/joc.2068) and more importantly compare different covariates commonly used in non-stationary frequency analysis of extreme rainfall in the context of Korea (see my comment below about the lack of description on how the covariate is selected and used). These two recommendations would increase the representativeness of the results but also provide regional insights for Korea. At the current state of the manuscript, the reader cannot know if these results are only valid for these two stations and with this covariate.

Response #1 :

Your detailed comments were very helpful in making a better manuscript. The authors would like to express great gratitude for this. First, let me tell you that some of the content in the discussion section has been moved to the methodology section. You can see this in the methodology section of the revised manuscript. Data from 11 sites, which began to be observed in 1961, were further analyzed. That is, a total of 13 sites were used in this study, including 2 sites that were previously applied. As a covariate, analysis was performed by adding surface air temperature in addition to the dew point temperature. The results of applying the added sites and an added covariate were prepared in the form of Supplementary Material and included in the revised manuscript. Also, as a figure showing the final result, Figure 7 of the revised manuscript was newly added. This further analysis may dispel concerns about whether the method proposed in this study applies only to two sites or is not valid only for dew point temperature. In addition, further analysis results will increase the representativeness of the results
derived from this study and provide local insights into Korea. As you mentioned, because of the high level of practical interest in non-stationary frequency analysis, the concept of "reference covariate" has been described in more detail in the methodology and discussion sections. You can find out about this in the Methodology section and the Discussion section. More specific details of how and where the manuscript has been revised are described in response to the comments presented below.

================================

Figure 7(a) shows the values of the negative log likelihood function of the stationary model and the non-stationary models at 13 sites. The stationary model, the SAT-based non-stationary model, and the DAT-based non-stationary model were found to have no significant difference in the fit performance with the observed POT excesses. Figure 7(b) shows the h-factor of rainfall quantile corresponding to the return level of 100-year. When all the values of covariate observed on the day of POT excesses are considered ("DPT" and "SAT" in Figure 7(b)), at all sites except Mokpo site, the non-stationary h-factor is greater than the stationary h-factor. However, when the reference covariate is applied, the non-stationary h-factor is smaller than the stationary h-factor. Results from 13 sites and most of the non-stationary models using SAT or DPT as a covariate indicate that how to determine the appropriate value of the covariate corresponding to the rainfall quantile plays an important role in securing the reliability of the non-stationary frequency analysis.

===================================

Comment #2 :

My first comment is about the title, quite long and not very informative of the main scientific results of the paper. Something like this title might be better: Uncertainty of non-stationary frequency analysis applied to extreme rainfall in Korea.

Response #2 :

Discussions with the authors are underway. The title so far determined is "Uncertainty in non-stationary frequency analysis of Korea's daily rainfall POT excesses associated with covariates". The results reflect the authors' opinion that the title is appropriate, including keywords such as POT, coviariate, and non-stationary. The current title can be revised through communication with researchers.

Comment #3 :

Abstract, line 28, this sentence it not clear, maybe too general: "However, since the parameters of the estimated probability distribution contain a lot of uncertainty". Abstract, line 40-42, this whole section below is quite trivial. Of course when a wrong covariate is selected in the POT model there is a stronger uncertainty. I don't see a major finding here. Overall the abstract needs a major upgrading to better present the main findings of the study.

Response #3 :

In order to avoid describing general contents and to better express the main results of the study, the abstract has been greatly upgraded as follows:

==============================

[revised manuscript text omitted]

==============================

Comment #6 :

Page 4, line 111, Why the authors are considering dew point temperature as a covariate for daily rainfall extremes for their stations in Korea ? Do previous works justify this choice ? No justification is given.

Response #6 :

Considering the dew point temperature as a covariate for daily rainfall extremes has been suggested in previous studies. We would like to establish the justification for selecting DPT or SAT as a covariate for rainfall extremes by briefly introducing two representative prior studies as follows:

============================

In this study, a non-stationary frequency analysis using dew point temperature (DPT) or surface air temperature (SAT) as a covariate is performed. To obtain a necessary understanding of the relationship between daily rainfall and DPT and daily rainfall and SAT in Korea, two prior studies have been conducted (Sim et al., 2019; Lee et al., 2020). Sim et al. (2019) analyzed the effects of DPT and SAT on daily rainfall extremes. Their results indicated that even if there was some cooling effect in the event of summer rainfall (Ali and Mishra, 2017), daily rainfall extremes in Korea were very sensitive to DPT and SAT. Lee et al. (2020) presented a procedure for performing non-stationary frequency analysis using DPT or SAT as a covariate. They revealed that non-stationary frequency analysis using future DPT or SAT may yield more reasonable and persuasive projections of future rainfall extremes. The purpose of this study is to focus on the uncertainty of covariate-based non-stationary frequency analysis using DPT or SAT.

============================

Comment #7 :

Page 5: line 130, "Daily rainfall depth of 0.1 mm or more was applied to the analysis," is not very clear. Does this mean you consider daily rainfall lower to 0.1 mm =0 ? Is this

related to rain gauges uncertainties ? How the results can be impacted by this choice?

Response #7 :

The Korea Meteorological Administration considers precipitation as more than 0.1 mm per day to be considered an official precipitation day. This is related to rainfall measurement equipment of the applied sites. In fact, including records with daily precipitation of 0.1 mm or less has no significant effect on results. A description of this has been added to the revised manuscript as follows:

===============================

Since the Korea Meteorological Administration only recognizes precipitation recorded at 0.1 mm or more per day as official precipitation, daily rainfall depth of 0.1 mm or more was applied to the analysis in this study. An example of this wet threshold can also be found in Chan et al. (2016). In fact, the application of a wet threshold does not significantly affect the results of quantile regression.

===============================

Comment #8 :

Page 7, line 170: I guess you mean here instead the "scale" parameter.

Response #8 :

Your comment is correct. The scale parameter is correct. Thank you for fixing it right away. You can see the correction as below:

===============================

Although studies considering the non-stationarity of the threshold of the POT series have been conducted (Tramblay et al., 2012), in this study, the non-stationarity was given only to the scale parameters of the GP distribution as follows (Um et al., 2017):

===============================

Comment #9 :

There is no indication on how the covariate is included in the model; is it the Dew point temperature the same day of the extreme rainfall event? at the starting day of a rainfall event or its peak ? On the opposite, is it computed for the week, or the months before the event ? No information is provided here.

Response #9 :

Covariate is defined as the daily average DPT or SAT on the day POT excesses occur. This information was included in the revised manuscript and described in the lower part of equation (4) as follows:

==================================

Eq. (4) tells how the covariate DPT or SAT is included in the model. The daily averaged DPT or SAT observed on the day of occurrence of each POT excess is included in the scale parameter of the GP distribution as shown in Eq. (4) to construct the non-stationary GP distribution. That is, when $\alpha\_2 > 0$, the larger the DPT or SAT, the larger the scale parameter.

==================================

Comment #10 :

Page 12, line 327, "However, under the condition that DPT is not given in advance" not clear to me. Do you mean you draw randomly DPT values instead of the values corresponding to the days with extreme rainfall?

Response #10 :

In "under the condition that DPT is not given in advance," our intention was to refer to the condition of extracting samples of the scale parameter $\alpha$ using all DPTs observed on the day the POT excess occurred. The manuscript has been revised as follows to convey clear meaning.

==================================

The h-factor of rainfall quantile corresponding to the return level of 100-year was calculated in two ways. First, under the condition that the reference DPT is given (i.e., when the reference value of DPT is applied), the h-factor of the non-stationary model is reduced by 37 % (at Busan site) and 28 % (at Seoul site) than that of the stationary model. However, under the condition that all observed DPTs corresponding to POT excesses are applied, the uncertainty from parameter estimation and the effects from extreme values of the covariate overlap, and the h-factor of the non-stationary model exceeds the h-factor of the stationary model. That is, if samples of the scale parameter (i.e., $\alpha$) is made by combining all samples of the coefficients of the scale parameter (i.e., $\alpha\_1$ and $\alpha\_2$) and samples of all observed DPTs corresponding to each POT excess, the uncertainty of rainfall quantiles in the non-stationary model is greater than the uncertainty of rainfall quantiles in the stationary model. The amplification of the uncertainty in the non-stationary model is because, as can be seen from Eq. (4), samples of some extreme DPTs significantly dissipate the samples of the scale parameter of the non-stationary GP distribution. This can also be confirmed through the lower right figure of Figure 4(a) and (b). The width of the 95PPU of the scale parameter of the non-stationary model corresponding to the value of the individual DPT is not significantly different from the width of the scale parameter of the stationary model. However, when all observed DPTs corresponding to the POT excesses are involved in sampling of the scale parameter, it can be recognized that the range of the 995 PPU of the scale parameter of the non-stationary model is very wide.

==================================

Comment #11 :

Page 13: the whole page/paragraph is quite long and not very clear to me, it could be shortened to the main findings.

Response #11 :

[Figure]

This section has been rewritten as follows, short and clearly centered on the main results.

========================================

We want to note here the condition in which the value of the covariate is given. In the upper left figure of Figure 4(a) and (b), the stationary quantile has a single value, while the ensemble average of the non-stationary quantile shows various values depending on the value of DPT. In addition, the 95 PPU of the stationary quantile has a constant range regardless of the value of the covariate, whereas the 95 PPU of the non-stationary quantile has a relatively wider range depending on the value of the covariate (see upper right figure in Figure 4(a) and (b)). This is due to the covariate dependence inherent in the scale parameter of the non-stationary GP distribution, as mentioned before. That is, since the range of the ensemble of the non-stationary rainfall quantile is a result of additionally reflecting the extreme values of the covariate in addition to the parameter uncertainty, it is more likely to be formed relatively wider than the range of the ensemble of the stationary rainfall quantile. It should be noted, however, that the width of the non-stationary 95PPU for a particular covariate value is less than the width of the stationary 95PPU. In fact, since the covariate corresponding to each POT excess is a known value, the h-factor of the rainfall quantile corresponding to each POT excess can be obtained (see lower left figure in Figure 4(a) and (b)). Given the value of covariate, it can be recognized that the non-stationary h-factor is smaller than the stationary h-factor. That is, if the value of the covariate of the non-stationary model can be determined, there is a room to say that the non-stationary frequency analysis is better in terms of reliability than the stationary frequency analysis.

========================================

Comment #12 :

Page 14, line 366, this equation should be in the Method section.

Response #12 :

The given equation and its description have been moved to the methodology section. It can be confirmed from the part where Eq. (5) of the revised manuscript is located.

Comment #13 :

The beginning of section 4.1 is obviously not a discussion and should be in the results section

Response #13 :

The beginning of section 4.1 has been moved to the results section. You can see the shifted content after Figure 4 of the revised manuscript.

Comment #14 :

Page 14, lines 386-390, this information should be in the method section, we should not discover in the discussion section how the covariate was implemented in the model. see my comment above.

Response #14 :

The section you mentioned has moved to the Methodology section. How the covariate was implemented in our model is described in the description of Eq. (4) of the revised manuscript. Specific modifications are already included in the answers to previous queries.

Comment #15 :

The issue of setting a reference covariate to a given return level is an interesting aspect. I believe it is necessary to first analyze the response of extreme rainfall to different values taken by the covariate, and as mentioned here it is difficult to identify a single value of the covariate related to a high risk of extreme rainfall. Yet this aspect needs more discussion, I don't see an added value of randomly selecting a covariate from a

pre-defined distribution (of the covariate).

Response #15 :

We also believe that the uncertainty analysis of randomly selecting a covariate from a predefined distribution of covariate is not feasible. The method of randomly selecting the covariate is implemented under the condition that all observed covariate samples corresponding to POT excesses are applied in this study. Therefore, this study investigated the response of covariate and rainfall quantile, and introduced the concept of reference covariate as an alternative. The first half of Section 4.1 of the revised manuscript was amended as follows:

===================================

[revised manuscript text omitted]

====================================

Comment #17 :

Conclusions, lines 498-501, these sentences are elements of context and should not appear here but rather in the introduction.

Response #17 :

In the revised manuscript, the sentences were moved to the beginning of the introduction.

Comment #18 :

Table 2: units?

Response #18 :

The numbers in Table 2 refer to m-factor and h-factor defined in Eqs. (9) and (10), respectively. The units of m-factor and h-factor are dimensionless. It was mentioned that it was dimensionless in the introduction of Eqs. (9) and (10). Also, what the numbers in Table 2 mean was more clearly described in the text referring to Table 2.

==================================

In this study, the following dimensionless quantitation factors were defined to quantify the uncertainty between the stationary and non-stationary models: Table 2 shows the results at Busan and Seoul sites. For reference, the results of applying DPT or SAT as a covariate at other sites are shown in Table S2 of the supplementary material.

==================================

Please also note the supplement to this comment:
https://www.hydrol-earth-syst-sci-discuss.net/hess-2020-167/hess-2020-167-AC1-supplement.zip

**dR/dDPT = 7.338% at $\tau$ = 0.95, Busan**

**Fig. 1.** Sensitivity of 95 % daily rainfall depth to dew-point temperature at (a) Busan and (b) Seoul sites

[Figure]

**Fig. 2.** Mean residual life plot at (a) Busan and (b) Seoul sites

[Figure]

Fig. 3. . Posterior distribution of parameters of stationary and non-stationary GP distribution. (a) Scale and (b) shape parameters at Busan site, and (c) scale and (d) shape parameters at Seoul site

[Figure]

**Fig. 4.** Changes in uncertainty for co-variate at (a) Busan and (b) Seoul sites

The stationary rainfall quantile is indicated as a blue vertical line since it is a single value. The non-stationary rainfall quantiles were calculated using the average of the parameter ensemble sampled by MCMC and the DPT observed on the day of POT excesses (red dotted line). In this figures, 'NS (3 °C up)' is an empirical distribution of rainfall quantile derived using DPTs that add 3 °C to DPTs observed on the day of POT excesses. Likewise, 'NS (5 °C up)' is an empirical distribution of rainfall quantile under the scenario condition where DPT has risen 5 °C due to global warming.

**Fig. 5.** Rainfall quantile estimates at (a) Busan, and (b) Seoul sites for return level of 100-year using observed dew-point temperature and global warming scenarios.

**Fig. 6.** Selection of reference dew-point temperature for estimating rainfall quantiles at (a) Busan and (b) Seoul sites.

Fig. 7. Performance of stationary and non-stationary frequency analysis models

[Figure]

**Fig. 8.** Effect of the number of samples on the uncertainty of rainfall quantile using reference dew-point temperature.

**likelihood of increase over change rate**

**Fig. 9.** . Likelihood of increase over change rate of rainfall quantile for return level of 100-year.

**Fig. 10.** Procedure for analyzing uncertainty in rate of change

[Figure]

---

## Referee Comment (RC2) · Anonymous Referee #2 · 30 Jun 2020

The current manuscript presents the nonstationary frequency analysis based on the POT precipitation data. The presented manuscript sounds interesting and contains the novelty. However, the assumption they made was not clearly explained and its still further explanation must be included. Therefore, I recommend that the current manuscript needs major revisions before publication. Detailed comments are attached.

L111 Specific information and references must be added to support the selection of DPF as a covariate. Physical relation must be also included between dpt and extreme rainfall.

[Figure]

L158-162 please make it italic and also for x throughout the paper. All the symbols must be italic unless matrix or vector.

L172 This one paragraph is not sufficient to set nonstationary model only for shape parameter. Detailed description must be included with more references.

L194 detailed description and references must be added to validate that these factors are meaningful.

L331-334 The sentence must be improved

L804 The range for (a) and (c) must be changed as shorter than 0-100. It seems that scale parameter has very accurate and small variance. However, in reality it is not.

L829 is y-axis 'realtive frequency' or pdf? 'realtive frequency'=ni/N whil pdf =ni/(N*dx). Check it.

L840 circle black line and blue line are not explained properly. F(DPT) does not seem to be empirical cumulative probabilities (see blue and red lines). It is just cumulative distribution function.

---

## Referee Comment (RC3) · Anonymous Referee #3 · 3 Jul 2020

The manuscript models the POT based extreme rainfall at Busan and Seoul sites of Korea using the Generalized Pareto distribution fitted under stationary and non-stationary settings. The authors compare the stationary GPD and non-stationary GPD based on the parameter uncertainty estimated using the Metropolis-Hastings (MH) algorithm. The manuscript can be published after addressing the following comments:

For constructing non-stationary GPD, the authors use DPT as the covariate. The reason for selecting DPT as a covariate is not clearly mentioned in the manuscript. Further, the authors should include a number of other covariates that affect the rainfall of the

study area in the non-stationary setting.

The reason estimating the parameters using the probability weighted moments (PWM) over the other state-of-the-art methods such as the maximum likelihood or L-moments should be mentioned in the manuscript.

The language of the manuscript is not adequate for an international journal. There are many vague/substandard sentences throughout the manuscript. For example, Line # 44-48, 75-79, etc. Further, the title of the manuscript is not clear and wordy. Include rainfall or precipitation in the title.

Fig. 2: Add legend or explain different lines in the figure caption.

Fig. 3: Why the PDF of the non-stationary model is shown for the DPT values of 20.2576 (Busan site) and 21.4962 (Seoul site)? Expand S and NS in the legend.

Most of the Figures & Tables: Use sentence case for figure title, legend and axis title.

---

## Author Comment (AC3) · 3 Jul 2020

Comment #1

The current manuscript presents the nonstationary frequency analysis based on the POT precipitation data. The presented manuscript sounds interesting and contains the novelty. However, the assumption they made was not clearly explained and its still further explanation must be included. Therefore, I recommend that the current manuscript needs major revisions before publication. Detailed comments are attached.

Your detailed comments were very helpful in making a better manuscript. The authors would like to express great gratitude for this. The main additions are as follows. First, data from 11 sites, which began to be observed in 1961, were further analyzed. That is, a total of 13 sites were used in this study, including 2 sites that were previously applied. As a covariate, analysis was performed by adding surface air temperature in addition to the dew point temperature. The results of applying the added sites and an added covariate were prepared in the form of Supplementary Material and included in the revised manuscript. Also, as a figure showing the final result, Figure 7 of the revised manuscript was newly added. This further analysis may dispel concerns about whether the method proposed in this study applies only to two sites or is not valid only for dew point temperature. In addition, further analysis results will increase the representativeness of the results derived from this study and provide local insights into Korea. More specific details of how and where the manuscript has been revised are described in response to the comments presented below.

––––––––––––––––––––––––––––––––––––––––––––––––––––––

Figure 7(a) shows the values of the negative log likelihood function of the stationary model and the non-stationary models at 13 sites. The stationary model, the SAT-based non-stationary model, and the DAT-based non-stationary model were found to have no significant difference in the fit performance with the observed POT excesses. Figure 7(b) shows the h-factor of rainfall quantile corresponding to the return level of 100-year. When all the values of covariate observed on the day of POT excesses are considered ("DPT" and "SAT" in Figure 7(b)), at all sites except Mokpo site, the non-stationary h-factor is greater than the stationary h-factor. However, when the reference covariate is applied, the non-stationary h-factor is smaller than the stationary h-factor. Results from 13 sites and most of the non-stationary models using SAT or DPT as a covariate indicate that how to determine the appropriate value of the covariate corresponding to the rainfall quantile plays an important role in securing the reliability of the non-stationary frequency analysis.

—————————————————————————————————————————-

Comment #2

[L111] Specific information and references must be added to support the selection of DPF as a covariate. Physical relation must be also included between dpt and extreme rainfall.

We will add information and related references that support DPT or SAT as the covariate to the revised manuscript below. We also included a description of the physical relationship between DPT or SAT and rainfall extreme.

—————————————————————————————————————————-

In this study, a non-stationary frequency analysis using dew point temperature (DPT) or surface air temperature (SAT) as a covariate is performed. As can be seen from Leopore et al. (2014), there is a strong scaling relationship between rainfall extreme and DPT or rainfall extreme and SAT. In addition, changes in DPT and SAT can directly affect the atmospheric moisture retention governed by the Clausius-Clapeyron equation, and in warmer climates, the moisture content of the atmosphere increases and the intensity of precipitation increases at a similar rate (Trenberth et. al., 2003; Giorgi et al., 2019). That is, according to the Clausius-Clapeyron relationship, the amount of moisture in the atmosphere increases exponentially as the temperature increases, and the amount of moisture in the atmosphere represents an increase rate of 6 - 7 %/K when other atmospheric conditions are kept constant. To obtain a necessary understanding of the relationship between daily rainfall and DPT and daily rainfall and SAT in Korea, two prior studies have been conducted (Sim et al., 2019; Lee et al., 2020). Sim et al. (2019) analyzed the effects of DPT and SAT on daily rainfall extremes. Their results indicated that even if there was some cooling effect in the event of summer rainfall (Ali and Mishra, 2017), daily rainfall extremes in Korea were very sensitive to DPT and SAT. Lee et al. (2020) presented a procedure for performing non-stationary frequency analysis using DPT or SAT as a covariate. They revealed that non-stationary

frequency analysis using future DPT or SAT could yield more reasonable and persuasive projections of future rainfall extremes. The purpose of this study is to focus on the uncertainty of covariate-based non-stationary frequency analysis using DPT or SAT. (Additional references) Ali, H. and Mishra, V. (2017) Contrasting response of rainfall extremes to increase in surface air and dewpoint temperatures at urban locations in India. Scientific Report, 7, 1228, DOI:10.1038/s41598-017-01306-1. Giorgi, F., Raffaele, F. and Coppola, E. (2019) The response of precipitation characteristics to global warming from climate projections. Earth System Dynamics, 10, pp. 73-89. Lepore, C., Veneziano, D. and Molini, A. (2014) Temperature and CAPE dependence of rainfall extremes in the eastern United States, Geophysical Research Letters, 42, pp. 74–83. Trenberth, K., Dai, A., Rasmussen, R. and Parsons, D. (2003) The changing character of precipitation. Bulletin of the American Meteorological Society, 84, pp. 1205-1218.

———————————————————————————————————-

Comment #3

[L158-162] please make it italic and also for x throughout the paper. All the symbols must be italic unless matrix or vector.

We will modify all the formulas in the text in italics.

Comment #4

[L172] This one paragraph is not sufficient to set nonstationary model only for shape parameter. Detailed description must be included with more references.

There is a typo. The parameter mentioned here is the scale parameter. We will accept the opinions of the referee and add the following explanation to the revised manuscript.

———————————————————————————————————-

In non-stationary frequency analysis, temporally changing parameters are applied to the probability distribution function (PDF). Various types of functions are applied to the

parameters that change over time. In general, the shape parameter is often set to constant (Lopez and Frances, 2013), but location or scale parameters are often considered functions of time or covariate. Ali and Mishra (2017) applied covariate to the location parameter of GEV, and Agilan and Umamahesh (2017) applied covariate to location and scale parameters of GEV. Non-stationary features in GP distribution are generally implemented by the scale parameter (Coles, 2001; Khaliq et al., 2006). Although non-stationarity can be expressed using the shape parameter, it is not a common practice since it is difficult to estimate the shape parameter, especially when considering covariates (Renard et al., 2006; Pujol et al., 2007). Although studies considering the non-stationarity of the threshold of the POT series have been conducted (Tramblay et al., 2012), in this study, the non-stationarity was given only to the scale parameters of the GP distribution as follows (Um et al., 2017): (Additional references) Khaliq, M., Ouarda, T., Ondo, J., Gachon, P. and Bobee, B. (2006) Frequency analysis of a sequence of dependent and/or non-stationary hydro-meteorological observations: A review. Journal of Hydrology, 329, pp. 534–552. Pujol, N., Neppel, L. and Sabatier, R. (2007) Regional tests for trend detection in maximum precipitation series in the French Mediterranean region. Hydrological Sciences Journal, 52, pp. 956–973. Renard, B., Lang, M. and Bois, P. (2006) Statistical analysis of extreme events in a nonstationary context via a Bayesian framework. Case study with peak-over-threshold data. Stochastic Environmental Research and Risk Assessment, 21, pp. 97–112.

———————————————————————————————————-

Comment #5

[L194] detailed description and references must be added to validate that these factors are meaningful.

It seems to be related to L294. We will explain the meaning of m-factor and h-factor in more detail, and add necessary references to the revised manuscript as follows:

———————————————————————————————————-

In fact, m-factor and h-factor can be seen as quantification of confidence intervals of ensembles simulated by MCMC. That is, the m-factor and h-factor of the estimated value indicate how accurate the estimate is or how much uncertainty is (Odura et al., 2020). The greater the uncertainty of the parameter or rainfall quantile, the greater the value of 95 PPU. That is, the quantitation factors of uncertainty expressed by m-factor and h-factor reflect the diffusion or lack of precision of the ensemble sampled from the posterior distribution (Motavita et al., 2019). (Additional reference) Motavita, D, Chow, R., Guthkea, A. and Nowaka, W. (2019) The comprehensive differential split-sample test: A stress-test for hydrological model robustness under climate variability. Journal of Hydrology, 573, pp. 501–515.

————————————————————————————————————-

Comment #6

[L331-334] The sentence must be improved.

We will revise the relevant sentences as follows:

————————————————————————————————————-

The above results indicate that although the non-stationary model is better in fitting performance for the observed samples, it is difficult to admit that the non-stationary model is more reliable than the stationary model due to the influence of extreme values of the covariate when estimating rainfall quantile. Ouarda et al (2020) also produced similar results using the annual maximum rainfall series and the non-stationary GEV distribution.

————————————————————————————————————-

Comment #7

L804 The range for (a) and (c) must be changed as shorter than 0-100. It seems that scale parameter has very accurate and small variance. However, in reality it is not.

The horizontal axis range in Figure 3 was set to visually indicate the range of prior-distribution. In fact, comparing the ensembles of scale and shape parameters, we can see that the scale parameters are relatively more accurate and show less coefficient of variation. For further explanation of this fact, we will add a description of the part corresponding to the revised manuscript as follows:
* * *
Table 1 shows the final estimated parameters at Busan and Seoul sites. The parameter estimation value of the MH algorithm was defined as the ensemble average of samples extracted by MCMC from the posterior distribution as mentioned in Eq. (7). The parentheses of the parameter estimation values by the MH algorithm in Table 1 are the coefficient of variation of the parameter. It can be found that the PWM and MH algorithms give similar parameter values for both scale and shape parameters. The negative logarithm likelihood (nllh) was also calculated similarly. From the above results, it can be recognized that when the POT series is to be fit to the GP distribution, parameter estimation by the MH algorithm is applicable, and information about the uncertainty of the estimated parameters is also obtainable. It can also be found that the coefficient of variation of the ensemble of scale parameters sampled by MCMC is less than 10 %, while the coefficient of variation of the ensemble of shape parameters is around 40 %. This means that the uncertainty of the shape parameters is relatively higher. Results for other sites tend to be similar to those obtained at Busan and Seoul sites. Results for other sites are shown in Table S1 of Supplemental Material.
* * *
Comment #8

[L829] is y-axis 'realtive frequency' or pdf? 'realtive frequency'=ni/N whil pdf =ni/(N*dx). Check it.

As a result of checking, the PDF is correct. We will replace the vertical axis label of

[Figure]

Figure 5 with PDF.

Comment #9

[L840] circle black line and blue line are not explained properly. F(DPT) does not seem to be empirical cumulative probabilities (see blue and red lines). It is just cumulative distribution function.

As in the reviewer's opinion, the F(PDF) of the original manuscript is correct. However, we will revise Figure 6 as shown below:

————————————————————————————————————————-

The formula for rainfall quantile X_T corresponding to the return level of T-year in the non-stationary GP distribution using covariate is as follows:

$X\_T = x\_o + 1/k\ e^{(\alpha\_1 + \alpha\_2 Z)} [1 - (1/\lambda T)^k]$. (5)

From Eq. (5), rainfall quantile X_T appears as a function of covariate Z. That is, Eq. (5) shows that various rainfall quantiles are calculated depending on the value of the covariate even at the same return level. Therefore, one of the problems to be solved in the non-stationary frequency analysis using a covariate is how to set the value of the covariate corresponding to a specific quantile. Since it is often required to have a single design rainfall depth in practice, it is very cumbersome to give a result of calculating rainfall quantiles of various values depending on a change in a covariate. From Eq. (5), the DPT value (i.e., reference DPT) of the non-stationary GP distribution that returns the rainfall quantile equal to the stationary GP distribution can be calculated (reference SAT can be calculated in the same way). Figure 6 shows an example of determining a reference DPT. The results of calculating the reference DPT at Busan and Seoul sites indicate that the reference DPT increases as the return level increases. The right figure in Figure 6(a) and (b) shows the histogram of DPT corresponding to POT excesses. The distribution of DPT is slightly distorted to the left. It can be found that the reference DPT corresponding to various return levels at Busan and Seoul sites is similar

to the location of the mode of the DPT distribution. This fact reveals that covariate values that deviate significantly from the reference covariate (i.e., some extreme values of the covariate) amplify the uncertainty of rainfall quantile from the non-stationary frequency analysis. From the results of regression analysis of rainfall quantile for various return levels and the corresponding reference DPT, the relationship of DPT = $18.8589RL^{0.01555}$ (where RL is the return level in year and the unit of DPT is °C) was obtained at Busan site. At Seoul site, a relationship of DPT = $19.8540RL^{0.01728}$ was obtained. The coefficient of determination of the regression analysis was 0.99 or higher at Busan and Seoul sites. From these results, the reference DPT corresponding to the return level of 100-year at Busan site could be applied to 20.2567 °C and Seoul site to 21.4958 °C. As shown in Figure 6 and Figures S3 and S4 of Supplementary Material, the value of the reference covariate is almost completely dependent on the return level. It should be noted that the return level and the reference covariate are proportional to each other at some sites, and are inversely proportional to other sites. This means that it is not easy to identify a single covariate value corresponding to a rainfall quantile. In this study, we tried to overcome the problem of random sampling of covariates by introducing the concept of reference covariate when estimating rainfall quantile estimation and its uncertainty from non-stationary frequency analysis based on covariate. From a practical point of view, how to set the value of the reference covariate may be an important research topic in the covariate-based non-stationary frequency analysis.

——————————————————————————————————————————-

Please also note the supplement to this comment:
https://www.hydrol-earth-syst-sci-discuss.net/hess-2020-167/hess-2020-167-AC3-supplement.zip

————————————————————————

---

## Author Comment (AC4) · 5 Jul 2020

Comment #1

The manuscript models the POT based extreme rainfall at Busan and Seoul sites of Korea using the Generalized Pareto distribution fitted under stationary and non-stationary settings. The authors compare the stationary GPD and non-stationary GPD based on the parameter uncertainty estimated using the Metropolis-Hastings (MH) algorithm. The manuscript can be published after addressing the following comments:

[Figure]

Your detailed comments were very helpful in making a better manuscript. The authors would like to express great gratitude for this. The main additions are as follows. First, data from 11 sites, which began to be observed in 1961, were further analyzed. That is, a total of 13 sites were used in this study, including 2 sites that were previously applied. As a covariate, analysis was performed by adding surface air temperature (SAT) in addition to the dew point temperature. The results of applying the added sites and an added covariate were prepared in the form of Supplementary Material and included in the revised manuscript. Also, as a figure showing the final result, Figure 7 of the revised manuscript was newly added. This further analysis may dispel concerns about whether the method proposed in this study applies only to two sites or is not valid only for dew point temperature. In addition, further analysis results will increase the representativeness of the results derived from this study and provide local insights into Korea. More specific details of how and where the manuscript has been revised are described in response to the comments presented below.

————————————————————————————————————————————

Figure 7(a) shows the values of the negative log likelihood function of the stationary model and the non-stationary models at 13 sites. The stationary model, the SAT-based non-stationary model, and the DAT-based non-stationary model were found to have no significant difference in the fit performance with the observed POT excesses. Figure 7(b) shows the h-factor of rainfall quantile corresponding to the return level of 100-year. When all the values of covariate observed on the day of POT excesses are considered ("DPT" and "SAT" in Figure 7(b)), at all sites except Mokpo site, the non-stationary h-factor is greater than the stationary h-factor. However, when the reference covariate is applied, the non-stationary h-factor is smaller than the stationary h-factor. Results from 13 sites and most of the non-stationary models using SAT or DPT as a covariate indicate that how to determine the appropriate value of the covariate corresponding to the rainfall quantile plays an important role in securing the reliability of the non-stationary frequency analysis.

—————————————————————————————————————————————

Comment #2

For constructing non-stationary GPD, the authors use DPT as the covariate. The reason for selecting DPT as a covariate is not clearly mentioned in the manuscript. Further, the authors should include a number of other covariates that affect the rainfall of the study area in the non-stationary setting.

In addition to using DPT as a covariate, SAT was added. Based on a study by Sim et al. (2019), this study selected DPT or SAT as the covariate on the day of POT excesses. However, in the previous study (Lee et al., 2020), DPT or SAT prior to t-day was selected as well as the day when POT excesses occurred. This study aimed to address how to explain the amplification of uncertainty that occurs when covariates are included in non-stationary frequency analysis, rather than the issue of deciding what covariates are suitable for POT excesses in Korea. The reason why we used DPT or SAT as a covariate to construct the non-stationary GPD is described below in the revised manuscript. We also included a description of the physical relationship between DPT or SAT and rainfall extreme to further support this.

—————————————————————————————————————————————

In this study, a non-stationary frequency analysis using dew point temperature (DPT) or surface air temperature (SAT) as a covariate is performed. As can be seen from Leopore et al. (2014), there is a strong scaling relationship between rainfall extreme and DPT or rainfall extreme and SAT. In addition, changes in DPT and SAT can directly affect the atmospheric moisture retention governed by the Clausius-Clapeyron equation, and in warmer climates, the moisture content of the atmosphere increases and the intensity of precipitation increases at a similar rate (Trenberth et. al., 2003; Giorgi et al., 2019). That is, according to the Clausius-Clapeyron relationship, the amount of moisture in the atmosphere increases exponentially as the temperature increases, and the amount of moisture in the atmosphere represents an increase rate of 6 - 7 %/K

when other atmospheric conditions are kept constant. To obtain a necessary understanding of the relationship between daily rainfall and DPT and daily rainfall and SAT in Korea, two prior studies have been conducted (Sim et al., 2019; Lee et al., 2020). Sim et al. (2019) analyzed the effects of DPT and SAT on daily rainfall extremes. Their results indicated that even if there was some cooling effect in the event of summer rainfall (Ali and Mishra, 2017), daily rainfall extremes in Korea were very sensitive to DPT and SAT. Lee et al. (2020) presented a procedure for performing non-stationary frequency analysis using DPT or SAT as a covariate. They revealed that non-stationary frequency analysis using future DPT or SAT could yield more reasonable and persuasive projections of future rainfall extremes. The purpose of this study is to focus on the uncertainty of covariate-based non-stationary frequency analysis using DPT or SAT.

(Additional references)

Ali, H. and Mishra, V. (2017) Contrasting response of rainfall extremes to increase in surface air and dewpoint temperatures at urban locations in India. Scientific Report, 7, 1228, DOI:10.1038/s41598-017-01306-1.

Giorgi, F., Raffaele, F. and Coppola, E. (2019) The response of precipitation characteristics to global warming from climate projections. Earth System Dynamics, 10, pp. 73-89.

Lepore, C., Veneziano, D. and Molini, A. (2014) Temperature and CAPE dependence of rainfall extremes in the eastern United States, Geophysical Research Letters, 42, pp. 74–83.

Trenberth, K., Dai, A., Rasmussen, R. and Parsons, D. (2003) The changing character of precipitation. Bulletin of the American Meteorological Society, 84, pp. 1205-1218.

––––––––––––––––––––––––––––––––––––––––––––––––––––––––––––

Comment #3

The reason estimating the parameters using the probability weighted moments (PWM)

over the other state-of-the-art methods such as the maximum likelihood or L-moments should be mentioned in the manuscript.

In this study, we will add the reason for estimating the parameter using PWM to the modified manuscript as follows:

————————————————————————————————————-

The parameters of the GP distribution were estimated using the method of probability weighted moments (PWM) and MH algorithm, respectively. Although maximum likelihood estimation is an efficient method, it does not clearly show efficiency even in samples larger than 500 (Smith, 1985). The method of moments is generally known to be reliable except when the shape parameter is less than -0.2. When the likelihood that the shape parameter is less than 0 is high, PWM estimation is recommended (Hosking and Wallis, 1987). Figure 3 shows the result of PWM parameter estimation and the posterior distribution of parameters by the MH algorithm at Busan and Seoul sites.

(Additional reference)

Smith, R. (1985) Maximum Likelihood Estimation in a Class of Nonregular Cases, Biometrika, 72, pp. 67-90.

————————————————————————————————————-

Comment #4

The language of the manuscript is not adequate for an international journal. There are many vague/substandard sentences throughout the manuscript. For example, Line # 44-48, 75-79, etc. Further, the title of the manuscript is not clear and wordy. Include rainfall or precipitation in the title.

We will check for ambiguous or non-standard sentences so that the language of the manuscript is suitable for international journals. First of all, we will modify the abstract and line #75-79 as below. There is an ongoing discussion about what to do with title.

The current discussion is below:
* * *
(Abstract including Line #44-48)

Several methods have been proposed to analyze the frequency of non-stationary anomalies. The applicability of the non-stationary frequency analysis has been mainly evaluated based on the agreement between the time series data and the applied probability distribution. However, since the uncertainty in the parameter estimate of the probability distribution is the main source of uncertainty in frequency analysis, the uncertainty in the correspondence between samples and probability distribution is inevitably large. In this study, an extreme rainfall frequency analysis is performed that fits the Peak-over-threshold series to the covariate-based non-stationary Generalized Pareto distribution. By quantitatively evaluating the uncertainty of daily rainfall quantile estimates at 13 sites of the Korea Meteorological Administration using the Bayesian approach, we tried to evaluate the applicability of the non-stationary frequency analysis with a focus on uncertainty. The results indicated that the inclusion of dew-point temperature (DPT) or surface air temperature (SAT) generally improved the goodness of fit of the model for the observed samples. The uncertainty of the estimated rainfall quantiles was evaluated by the confidence interval of the ensemble generated by the Markov chain Monte Carlo. The results showed that the width of the confidence interval of quantiles could be greatly amplified due to extreme values of the covariate. In order to compensate for the weakness of the non-stationary model exposed by uncertainty, a method of specifying a reference value of a covariate corresponding to a non-exceedance probability has been proposed. The results of the study revealed that the reference co-variate plays an important role in the reliability of the non-stationary model. In addition, when the reference co-variate was given, it was confirmed that the uncertainty reduction of quantile estimates for the increase in the sample size was more pronounced in the non-stationary model. Finally, it was discussed how information on global temperature rise could be integrated with DPT or SAT-based non-stationary frequency analysis. It has been formulated how to quantify the uncertainty of the rate of change in future quantile due to global warming using rainfall quantile ensembles obtained in the uncertainty analysis process.

(Line #75-79)

Several methods have been proposed to address non-stationarities in the time series (Cunha et al., 2011; Yilmaz et al., 2013; Jang et al., 2015, Moon et al., 2016), and many studies have been conducted to examine changes in design rainfall depth or return levels under non-stationary conditions (Salvadori and DeMichele, 2010; Graler et al., 2013; Hassanzadeh et al., 2013; Salas and Obeysekera, 2013; Shin et al., 2014; Choi et al ., 2019).

(Title) Uncertainty in non-stationary frequency analysis of Korea's daily rainfall POT excesses associated with covariates

————————————————————————————————————-

Comment #5

Fig. 2: Add legend or explain different lines in the figure caption.

I will add the following description to the caption in Figure 2.

————————————————————————————————————-

Figure 2. Mean residual life plot at (a) Busan and (b) Seoul sites. The solid line is the mean of the excesses of the threshold, and the dotted line is approximated 95% confidence intervals.

————————————————————————————————————-

Comment #6

Fig. 3: Why the PDF of the non-stationary model is shown for the DPT values of 20.2576 (Busan site) and 21.4962 (Seoul site)? Expand S and NS in the legend.

The scale parameter in this study is a function of the covariate. Therefore, the posterior distribution of scale parameters depends on the covariate. The dependence of scale parameters on covariates is the most important part of this study. Since the uncertainty of the non-stationary model is excessively amplified when the dependence of the covariate is reflected in the uncertainty, this study attempted to prevent excessive amplification of the uncertainty of the non-stationary model by introducing the concept of a reference covariate. Through this, it was possible to secure the reliability of the non-stationary model. The 'S' and 'NS' in the legend in Figure 3 will be modified to 'Stationary' and 'Non-stationary', respectively. The revised manuscript is presented below:
* * *

[revised manuscript text omitted]

————————————————————————————————————–

Comment #7

Most of the Figures & Tables: Use sentence case for figure title, legend and axis title.

We will use the sentence case to modify the picture and table.

Please also note the supplement to this comment:
https://www.hydrol-earth-syst-sci-discuss.net/hess-2020-167/hess-2020-167-AC4-supplement.zip